# Solvent-free microwave synthesis of ultra-small Ru-Mo$_2$C@CNT with strong metal-support interaction for industrial hydrogen evolution

Xueke Wu[1], Zuochao Wang[1], Dan Zhang[1,2], Yingnan Qin[1], Minghui Wang[1], Yi Han[1], Tianrong Zhan[1], Bo Yang[2], Shaoxiang Li[2], Jianping Lai [1✉] & Lei Wang [1,2✉]

Exploring a simple, fast, solvent-free synthetic method for large-scale preparation of cheap, highly active electrocatalysts for industrial hydrogen evolution reaction is one of the most promising work today. In this work, a simple, fast and solvent-free microwave pyrolysis method is used to synthesize ultra-small (3.5 nm) Ru-Mo$_2$C@CNT catalyst with heterogeneous structure and strong metal-support interaction in one step. The Ru-Mo$_2$C@CNT catalyst only exhibits an overpotential of 15 mV at a current density of 10 mA cm$^{-2}$, and exhibits a large turnover frequency value up to 21.9 s$^{-1}$ under an overpotential of 100 mV in 1.0 M KOH. In addition, this catalyst can reach high current densities of 500 mA cm$^{-2}$ and 1000 mA cm$^{-2}$ at low overpotentials of 56 mV and 78 mV respectively, and it displays high stability of 1000 h. This work provides a feasible way for the reasonable design of other large-scale production catalysts.

[1] Key Laboratory of Eco-chemical Engineering, Key Laboratory of Optic-electric Sensing and Analytical Chemistry of Life Science, Taishan Scholar Advantage and Characteristic Discipline Team of Eco Chemical Process and Technology, College of Chemistry and Molecular Engineering, Qingdao University of Science and Technology, Qingdao, P. R. China. [2] Shandong Engineering Research Center for Marine Environment Corrosion and Safety Protection, College of Environment and Safety Engineering, Qingdao University of Science and Technology, Qingdao, P. R. China. ✉email: jplai@qust.edu.cn; inorchemwl@126.com

Environmental pollution and energy shortage have become two of the most serious problems facing today's society[1]. As an ideal clean energy and an important chemical raw material, hydrogen has received extensive attention from all over the world[2,3]. Electrocatalytic water splitting is an important means to produce hydrogen on a large scale and at a low cost, and it is also considered to be one of the potential strategies to solve these two major social problems[4–6]. Reasonable design of efficient and stable catalysts is a major problem in the current electrocatalytic hydrogen evolution reaction (HER)[7–9]. At present, platinum (Pt)-based catalyst is the best hydrogen evolution catalyst, but the low natural content of Pt and its high price limit its large-scale production[10–12]. The design and development of high-performance and low-cost catalysts with low precious metal loading catalysts have become the top priority in this field[13–15]. Ruthenium (Ru), due to its low cost (only 1/3 of the price of Pt) and high activity (the Gibbs free energy ($\Delta G_H$) of Ru–H bond is very close to the free energy of Pt–H bond in the center of HER volcanic map), so it has become one of the cheap substitutes for Pt[16,17]. Therefore, the reasonable design and large-scale preparation of Ru-based catalysts with low cost, high activity and high stability is one of the key scientific issues to realize low-cost electrolysis of water for hydrogen production[18,19].

Up to now, researchers have taken a series of measures to design Ru-based catalysts that can be used for production of $H_2$. On the one hand, the cost of preparing the catalyst can be reduced by reducing the content of Ru and improving the preparation method[20,21]. In the former, the addition of transition metal elements is usually used to reduce the content of Ru[22,23]. For the latter preparation method, the traditional synthesis methods of Ru-based catalyst (such as hydrothermal method[24,25], high-temperature calcination method[26], electrochemical deposition method[27], etc.) are not only time-consuming, highly equipment demanding, easy to form large nanoparticles, but also low in yield and difficult to mass production[28,29]. Moreover, in non-traditional heating techniques such as high-temperature pulse method[30], etc., although it can quickly prepare catalysts, it has high requirements on equipment and the process is difficult to control[31]. In addition, the synthesis process generally involves the participation of solvents, which consumes high energy and easily causes pollution[32,33]. Therefore, it is still a huge challenge to use a simple, fast and solvent-free synthesis method to prepare small-sized, low-ruthenium-loaded catalysts that can be mass-produced[34].

On the other hand, the catalytic activity of Ru-based materials can be improved mainly from the following two key factors: (1) to increase the number of accessible active sites of catalysts, which is usually achieved by increasing the specific surface area of the catalyst materials[35,36]. For example, the loading of small-sized Ru-based materials on carbon nanotubes can not only fully expose the active sites of the Ru-based materials, but also increase the electrical conductivity of materials[37]. (2) To enhance the intrinsic activity of the material, it is generally achieved by constructing multiple active sites (adopting the methods of element doping[38–40], constructing heterogeneous structure[41], surface modification[42], etc.). However, nanoparticles with multiple active sites synthesized by conventional methods usually have a large size (generally greater than 10 nm)[43]. While the inherent synergy between multiple active sites can promote the dissociation of $H_2O$ and coordinate the adjustment of the free energy of adsorption and desorption of the reaction intermediates and products to facilitate the reaction[44], but large-sized particles will produce unexpectedly long reaction paths, which will lead to undesirable transport and reaction resistance[45]. Therefore, what method should be adopted to comprehensively control small-sized Ru-based catalysts to shorten the distance between multiple active

sites and synergistically improve its catalytic activity is still a difficult problem[46].

Finally, the stability of the catalyst can be enhanced by constructing a strong metal-support interaction (SMSI)[47]. SMSI can not only prevent the agglomeration of the loaded nanoparticles and promote the electron transfer process between the carrier and the load, but also make the load firmly anchored on the carrier and greatly increase the stability of the material[48]. In addition to common metal oxide supports, low cost, controllable carbon materials can also be used as supports, but so far, there are few studies on such materials[49,50].

To sum up, although there are many design methods for Ru-based catalysts, it is still an arduous challenge to simultaneously reduce costs, increase activity, and enhance stability to realize the industrialized production of catalysts for $H_2$ production[51,52].

In this work, Ru-$M_xC$@CNT (M = Mo, Co, Cr) was synthesized in one step using a simple, fast and solvent-free microwave pyrolysis method. The acid treated carbon nanotubes were mixed with metal carbonyl salts ($Ru_3(CO)_{12}$, $M(CO)_x$) at room temperature. Under the rapid action of microwave radiation, the carbonyl salts decomposed to produce CO gas, which rapidly reduced metal ions to nanoparticles and had strong interaction with CNT support. This whole synthesis process only requires a simple step, no solvent is involved, and it is very fast, solvent-free and high yield. A series of physical characterizations and experimental results show that at rapid high temperature, Ru and $M_xC$ form a Ru-$M_xC$ heterojunction, and bond with the carbon element on the multi-wall carbon nanotube (MWCNT) to form a strong metal-support interaction, and the size of the formed heterojunction is only 3.5 nm, which improve the electrochemistry hydrogen evolution performance and increase structural stability. Among them, the Ru-$Mo_2C$@CNT catalyst shows superior performance to commercial Pt/C in 1.0 M KOH electrolyte. The catalyst only shows an overpotential of 15 mV at a current density of 10 mA cm$^{-2}$, and its Tafel slope is only 26 mV dec$^{-1}$. The catalyst Ru-$Mo_2C$@CNT has a superior intrinsic activity, which turnover frequency (TOF) value reaches 21.9 s$^{-1}$ at an overpotential of 100 mV. In addition, this catalyst can reach high current densities of 500 mA cm$^{-2}$ and 1000 mA cm$^{-2}$ at low overpotentials of 56 and 78 mV, respectively. At the same time, this catalyst has high electrochemical stability and structural stability. After 1000 h long-term electrochemical test, the current density has basically not attenuated, and the structure has also not changed.

## Results

### Synthesis and characterizations of Ru-$Mo_2C$@CNT catalyst.
The Ru-$M_xC$@CNT (M = Mo, Co, Cr) catalyst with a strong metal-support interaction was synthesized by a simple microwave pyrolysis method using acidified multi-walled carbon nanotubes (MWCNT) (Supplementary Figs. 1, 2) as the carrier, and only needed to be reacted at a high temperature in a household microwave oven for 100 s (Supplementary Fig. 3). Using MWCNT with high specific surface area, high mechanical strength and high conductivity as a carrier, the active components are highly and uniformly dispersed, which can increase the number of active sites accessible to the catalyst. In order to confirm the optimal metal to support ratio, we explored the HER catalytic activity of catalysts with different Ru-$Mo_2C$:CNT ratios. As shown in Supplementary Fig. 4, when the ratio of metal to support is 1:1, the catalytic performance is the best. Based on the 1:1 ratio of metal and support, the best content ratio of Ru: $Mo_2C$ was explored. Among them, the initial reactants $Ru_3(CO)_{12}$ and $Mo(CO)_6$ with mass ratios of 1:2, 1:1, and 2:1 were placed in a microwave oven for reaction. Different ratios of samples were

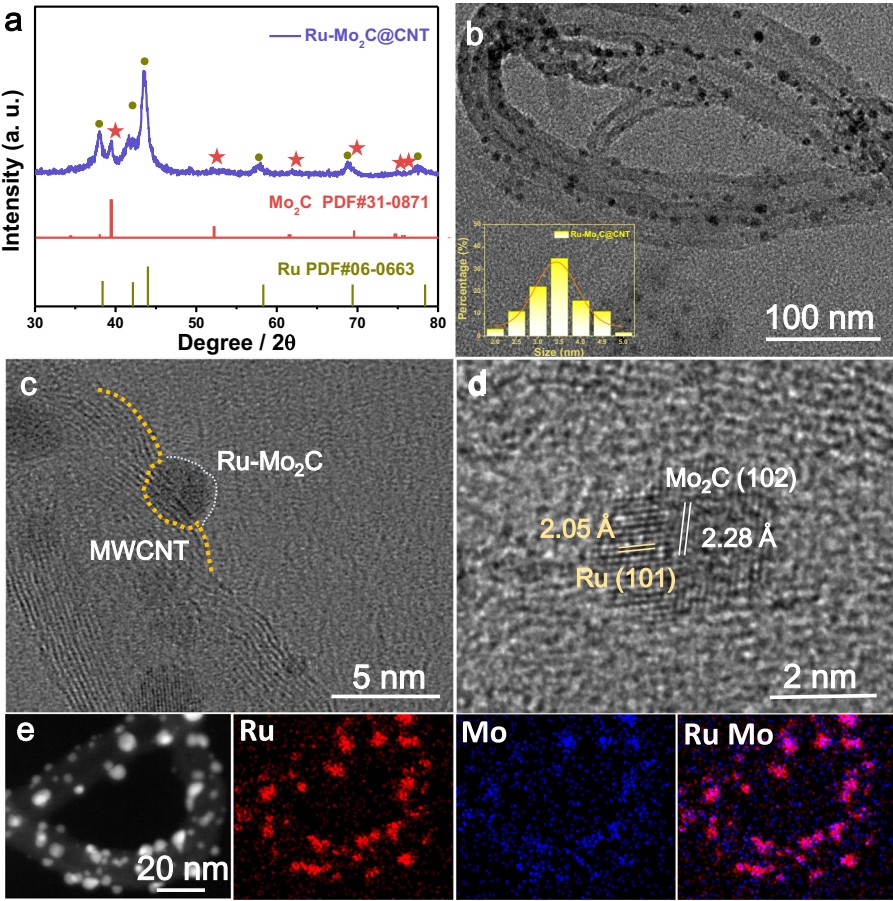

**Fig. 1 Physical characterization of Ru-Mo₂C@CNT catalyst. a** The XRD pattern of Ru-Mo₂C@CNT. **b** TEM image of Ru-Mo₂C@CNT, the inset is the size distribution of nanoparticles. **c, d** HRTEM images of Ru-Mo₂C@CNT. **e** HAADF-STEM image and corresponding EDX maps of Ru-Mo₂C@CNT for Ru, Mo, and Ru+Mo, respectively.

tested by inductively coupled plasma atomic emission spectrometer (ICP-AES) to obtain the actual ratio of Ru:Mo element content after the reaction was completed (Supplementary Table 1). Electrochemical hydrogen evolution tests were performed on three samples of different proportions in 1.0 M KOH solution, and we found that the electrochemical performance was the best when the Ru:Mo₂C element content ratio was 2:1 (Supplementary Fig. 5). And it was estimated by ICP-AES results that the loading of Ru-Mo₂C in the catalyst Ru-Mo₂C@CNT is about 19 wt%. Figure 1a showed the X-ray diffraction (XRD) pattern of Ru-Mo₂C@CNT. Among them, there are four obvious characteristic peaks at 39.5°, 52.1°, 61.6°, and 69.5°, which correspond to the (102), (221), (321) and (023) crystal planes of Mo₂C (PDF#31-0871) substances[53]. The characteristic peaks at 38.4°, 42.2°, 44.0°, 58.3°, 69.4°, and 78.4° belong to the (100), (002), (101), (102), (110), and (103) crystal planes of ruthenium (PDF#06-0663), respectively[51]. The morphology and structure of the catalyst Ru-Mo₂C@CNT were first characterized by scanning electron microscope (SEM) and transmission electron microscope (TEM). As shown in Fig. 1b and Supplementary Fig. 6, the nanoparticles were uniformly anchored on the CNT, with an average size of about 3.5 nm. According to Fig. 1c, it was found that the active component loaded on the CNT formed carbon layer cladding under the action of microwave heat radiation, and it was preliminarily believed that SMSI might be generated between the active component and the carrier CNT. The high-resolution transmission electron microscope (HRTEM, Fig. 1d) image showed the more intuitive structural features of the catalyst Ru-Mo₂C@CNT. The obvious interface between Ru and Mo₂C

can be observed from the image, which proved the typical heterojunction structure of Ru-Mo₂C. The d-spacing were 2.05 Å and 2.28 Å, corresponding to the (101) crystal plane of Ru and the (102) crystal plane of Mo₂C, respectively, which were very consistent with the XRD results. In addition, the energy-dispersive X-ray spectroscopy (EDX) element mapping images (Fig. 1e and Supplementary Fig. 7) showed that Ru and Mo elements are uniformly distributed on the CNT.

In order to deeply explore the valence state and unique electronic structure of Ru-Mo₂C@CNT catalyst, X-ray photoelectron spectroscopy (XPS) was used to test this catalyst and the comparative samples Ru@CNT and Mo₂C@CNT. First, a series of physical characterizations such as SEM, TEM, XRD, and XPS proved the successful preparation of comparative samples Ru@CNT (Supplementary Figs. 8–10) and Mo₂C@CNT (Supplementary Figs. 11–13). The existence of Ru, Mo and C elements was observed in Supplementary Fig. 14, which further confirmed the successful preparation of Ru-Mo₂C@CNT catalyst. Figure 2a shows the XPS spectrum of C 1 s, where the peaks of binding energy (BE) of 284.5 and 286.5 eV correspond to C=C and C=O, respectively. Compared with the single CNT material, its peak position migrated to a higher BE, indicating that under the action of microwave heat radiation, charge transfer occurred between the heterojunction particles and the carrier, resulting in SMSI phenomenon, in which electrons were transferred from the CNT carrier to the Ru-Mo₂C interface. According to Fig. 2b, the signal peak of C 1 s is partially coincident with the signal peak of Ru 3d. The peaks at 280.5 and 285.7 eV are metal Ru, corresponding to Ru $3d_{5/2}$ and Ru $3d_{3/2}$, respectively, while the peak at 281.0 eV

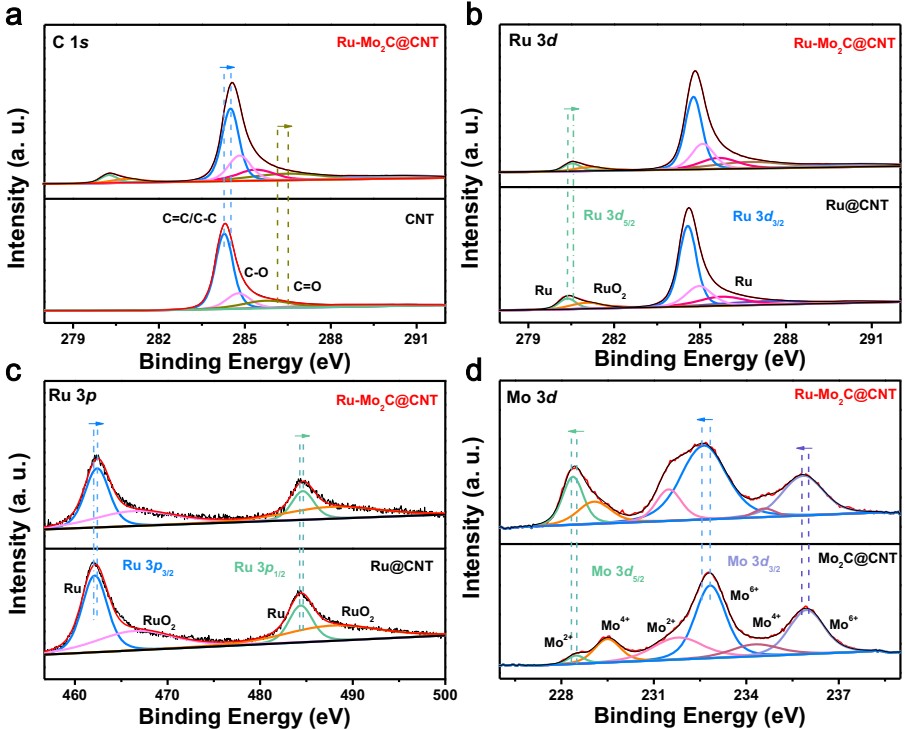

**Fig. 2 High-resolution XPS spectra of Ru-Mo$_2$C@CNT. a** High-resolution XPS spectrum of C 1$s$ of CNT and Ru-Mo$_2$C@CNT. **b** High-resolution XPS spectrum of Ru 3$d$ spectra of Ru@CNT and Ru-Mo$_2$C@CNT. **c** High-resolution XPS spectrum of Ru 3$p$ of Ru@CNT and Ru-Mo$_2$C@CNT. **d** High-resolution XPS spectrum of Mo 3$d$ of Mo$_2$C@CNT and Ru-Mo$_2$C@CNT.

belongs to RuO$_2$. The valence distribution of Ru was further observed from the deconvolution peak of Ru 3$p$ (Fig. 2c). The spectrum showed two peaks at about 462.6 and 467.6 eV, corresponding to the metallic ruthenium and ruthenium dioxide of Ru 3$p_{3/2}$. Correspondingly, the two signal peaks of metallic ruthenium and ruthenium dioxide were attributed to Ru 3$p_{1/2}$ appear at 484.9 and 488.9 eV, respectively[51]. The presence of ruthenium dioxide may be caused by slight oxidation of the sample when exposed to air. In addition, the above two figures also showed the changes of the binding energy of Ru element in the catalyst Ru-Mo$_2$C@CNT compared with Ru@CNT. Obviously, after the introduction of Mo element, both Ru 3$p$ and Ru 3$d$ binding energies moved towards higher binding energies, indicating that the introduction of Mo element caused the charge transfer between Ru and Mo$_2$C components. Figure 2d showed the XPS spectrum of Mo 3$d$ in the Ru-Mo$_2$C@CNT sample. Among them, the signal peaks at 228.4, 229.1, 231.4, and 234.5 eV attributed to Mo$^{\delta+}$ ($0 < \delta < 4$) species are usually Mo$_2$C substances. Since Mo$_2$C was easily oxidized on the surface, the characteristic peaks of Mo$^{6+}$ at 232.6 eV and 235.7 eV can also be observed. In addition, signal peaks attributed to metallic Mo were not detected[54]. Compared with Mo$_2$C@CNT, it can be clearly observed that the peak position of Mo element in the catalyst Ru-Mo$_2$C@CNT has also changed. The XPS analysis showed that there is a charge transfer between the carrier CNT and the heterojunction Ru-Mo$_2$C, which further proves the existence of the SMSI effect. Moreover, the Ru and Mo peaks between the two components of the heterojunction also shifted, indicating the existence of strong electron interaction between Ru and Mo$_2$C components. Combining the following experiments proved that all these contribute to the improvement of catalytic performance.

In addition, the defects of Ru@CNT, Mo$_2$C@CNT and Ru-Mo$_2$C@CNT were further explored through Raman spectroscopy. It can be seen from Supplementary Fig. 15 that the intensity ratio

of the D band and the G band ($I_D/I_G$) of the catalyst Ru-Mo$_2$C@CNT is 1.45, which was significantly higher than the intensity ratio of the control samples Ru@CNT (0.91) and Mo$_2$C@CNT (1.23). This indicated that the formation of the Ru-Mo$_2$C heterojunction in the microwave process induces more defects on the CNT, which facilitated the utilization of more active sites and enhances the performance of HER.

**Electrocatalytic performance tests toward HER.** The electrochemical HER performance of Ru-Mo$_2$C@CNT (catalyst loading is 0.14 mg cm$^{-2}$, equals to a Ru-Mo$_2$C loading of ca. 0.03 mg cm$^{-2}$) was evaluated in 1.0 M KOH solution saturated with N$_2$. For comparison, the HER performance of CNT, Mo$_2$C@CNT, Ru@CNT, and commercial Pt/C were measured under the same test conditions, which proved that the best electrocatalytic activity mainly comes from the formation of Ru-Mo$_2$C heterojunction. The linear sweep voltammetric (LSV) curves of the five electrocatalysts tested in 1.0 M KOH solution were shown in Fig. 3a. Compared with bare CNT which have basically no electrocatalytic activity, the HER activity of other catalysts increases in the following order: Mo$_2$C@CNT < Ru@CNT < Pt/C < Ru-Mo$_2$C@CNT. As shown in Fig. 3c, it is worth noting that Ru-Mo$_2$C@CNT only needs a low overpotential of 15 mV to reach the current density of 10 mA cm$^{-2}$, which is nearly 48 mV lower than Ru@CNT (63 mV) and about 309 mV lower than Mo$_2$C@CNT (324 mV). Correspondingly, Ru-Mo$_2$C@CNT catalyst showed lower overpotential than commercial Pt/C (33 mV), which further indicated that Ru-Mo$_2$C@CNT had a best electrocatalytic HER activity. In addition, we further compared the overpotentials of the materials of Mo$_2$C@CNT, Ru@CNT, Pt/C and Ru-Mo$_2$C@CNT at a current density of 20 mA cm$^{-2}$, and the results showed that the overpotentials of the four samples exhibited a significant decrease trend (Supplementary Fig. 16). A lower Tafel slope is also an important indicator for the performance of

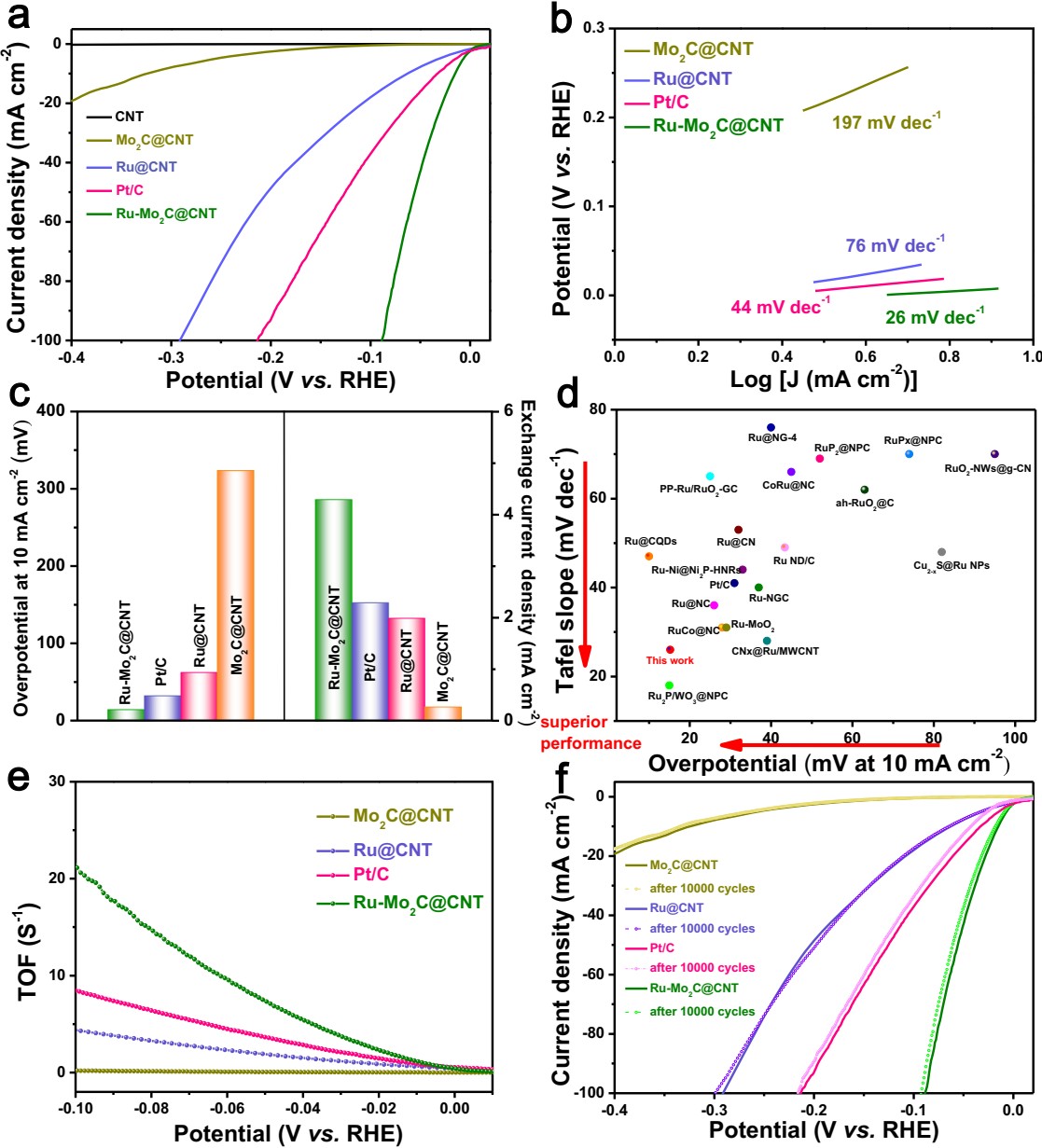

**Fig. 3 Electrocatalytic HER performance test of catalysts in N₂-saturated 1.0 M KOH solution. a** HER polarization curves of CNT, Mo₂C@CNT, Ru@CNT, Pt/C, and Ru-Mo₂C@CNT catalysts. **b** Tafel plots obtained from the polarization curves in **a**. **c** Comparison of overpotential changes at 10 mA cm⁻² and exchange current density. **d** In 1.0 M KOH solution, compared with the recently reported HER catalyst Tafel slope and overpotential at 10 mA cm⁻². **e** The potential-dependent TOF curves of Mo₂C@CNT, Ru@CNT, Pt/C, and Ru-Mo₂C@CNT. **f** Polarization curves for Mo₂C@CNT, Ru@CNT, Pt/C, and Ru-Mo₂C@CNT catalysts before and after 10,000 cycles.

electrocatalysis. In 1.0 M KOH solution, Ru-Mo₂C@CNT only showed a low Tafel slope of 26 mV dec⁻¹ (Fig. 3b), which was much lower than Pt/C (44 mV dec⁻¹), Ru@CNT (76 mV dec⁻¹) and Mo₂C@CNT (197 mV dec⁻¹), which implies the Volmer-Tafel mechanism as the HER pathway, in which recombination of chemisorbed hydrogen atoms is the rate-determining step. Exchange current density ($J_0$) is another important parameter reflecting translation kinetics, which can provide the internal electron transfer rate between the electrode and the catalyst surface[55]. The exchange current density ($J_0$) of the catalyst was extracted from the linear fitting of the micropolarization region (−10 to 10 mV)[56]. The exchange current density of Ru-Mo₂C@CNT was 4.3 mA cm⁻² (Fig. 3c and Supplementary Fig. 17b), which was better than

commercial Pt/C (2.3 mA cm⁻²) (Fig. 3c and Supplementary Fig. 18b). The exchange current densities of Ru@CNT and Mo₂C@CNT were 2.0 and 0.28 mA cm⁻², respectively, (Fig. 3c), which were slightly lower than the $J_0$ of the Pt/C catalyst. These values are in good agreement with the fitting results of the Butler-Volmer equation in the Tafel region (Supplementary Fig. 17a and Supplementary Fig. 18a). Obviously, the intrinsic catalytic activity of Ru-Mo₂C@CNT with a small-sized Ru-Mo₂C heterostructure is the best, and it exhibits the best HER performance in alkaline media[57,58]. From Fig. 3d, it was clear that the HER activity of our catalyst was better than most precious metal catalysts (including Pt-based and Ru-based HER catalysts) and non-precious metal catalysts. In order to evaluate the electrochemical surface area (ECSA)

of the catalyst, low potential copper deposition (Cu-UPD) was performed on the Ru-Mo$_2$C@CNT catalyst (Supplementary Fig. 19). The ECSA of Ru-Mo$_2$C@CNT is 97.6 m$^2$g$^{-1}$$_{Ru}$, which is larger than Pt/C (73.8 m$^2$g$^{-1}$), Ru@CNT (69.1 m$^2$g$^{-1}$) and Mo$_2$C@CNT (42.8 m$^2$g$^{-1}$) (Supplementary Fig. 20). This is because the formation of the Ru-Mo$_2$C heterojunction increases the active specific surface area of the catalyst compared with the single active site of Ru and Mo$_2$C. When evaluating HER electrocatalysts, the TOF value and the overpotential at 10 mA cm$^{-2}$ respectively reveal the intrinsic activity of the catalyst and the potential for practical applications. According to the estimated number of active sites, the TOF value of each active site of Ru-Mo$_2$C@CNT, Pt/C, Ru@CNT and Mo$_2$C@CNT in alkaline electrolyte was calculated (Supplementary Fig. 21). As shown in Fig. 3e, the catalyst Ru-Mo$_2$C@CNT exhibited a larger TOF of 21.9 s$^{-1}$ under an overpotential of 100 mV, which showed an activity better than most of the catalysts reported in the literature (Supplementary Table 2). In addition, the double-layer capacitance (C$_{dl}$) was used to further evaluate the active surface area of the material, which was calculated by measuring the CV curve in the potential range of 0.86 to 0.96 V (vs RHE, reversible hydrogen electrode) (Supplementary Fig. 21a-c). In 1.0 M KOH solution, the capacitance (C$_{dl}$) of Ru-Mo$_2$C@CNT is 17.2 mF cm$^{-2}$, which was much higher than Ru@CNT (9.5 mF cm$^{-2}$) and Mo$_2$C@CNT (7.2 mF cm$^{-2}$), and this result was consistent with the conclusion drawn by the Cu-UPD method (Supplementary Fig. 22). The electrochemical impedance spectroscopy (EIS) fitting results showed that the charge transfer resistance of Ru-Mo$_2$C@CNT (21.3 Ω) is smaller than that of Ru@CNT (26.0 Ω) and Mo$_2$C@CNT (30.6 Ω) (Supplementary Fig. 23). The results showed that the formation of Ru-Mo$_2$C heterostructure and the presence of SMSI effect improved the conductivity and interfacial electron transport capacity of the catalyst, increased the number of accessible active sites, and improved the electrocatalytic performance.

Ru-Mo$_2$C@CNT catalyst was successfully prepared by solvent-free microwave pyrolysis and its synthesis method is fast, simple and high in yield. Moreover, the catalyst had heterojunction structure and SMSI effect at the same time, showing high hydrogen evolution performance in 1.0 M KOH solution. In order to further prove the reliability and universality of the microwave pyrolysis method, the samples Ru-Co$_3$C@CNT and Ru-Cr$_{23}$C$_6$@CNT were successfully synthesized under the same conditions. The SEM images, TEM images, XRD patterns, and XPS spectra shown in Supplementary Figs. 24−2 indicate the successful preparation of the catalysts Ru-Co$_3$C@CNT and Ru-Cr$_{23}$C$_6$@CNT. Under the same operating environment and test conditions, the HER performance of Ru-Co$_3$C@CNT and Ru-Cr$_{23}$C$_6$@CNT were tested in 1.0 M KOH solution. Supplementary Figure 30a showed the LSV curves of these three catalysts. The LSV curve showed that the three electrocatalysts prepared by microwave solid-phase pyrolysis all have high electrochemical HER activity. The Ru-Co$_3$C@CNT and Ru-Cr$_{23}$C$_6$@CNT catalysts only need 32 mV and 27 mV overpotentials to reach the current density of 10 mA cm$^{-2}$ (Supplementary Fig. 30c). Moreover, only 57 and 56 mV overpotentials are required for Ru-Co$_3$C@CNT and Ru-Cr$_{23}$C$_6$@CNT to achieve the current density of 20 mA cm$^{-2}$ (Supplementary Fig. 31). In addition, the Tafel slopes of Ru-Co$_3$C@CNT and Ru-Cr$_{23}$C$_6$@CNT catalysts are 56 and 53 mV dec$^{-1}$, respectively, (Supplementary Fig. 30b). The low Tafel slope indicates that this kind of catalyst prepared by microwave pyrolysis can effectively accelerate HER dynamics. The exchange current densities of Ru-Co$_3$C@CNT and Ru-Cr$_{23}$C$_6$@CNT are also studied from linear fitting of micropolarization regions, which are 2.7 and 3.6 mA cm$^{-2}$, respectively, (Supplementary Fig. 30c). In order to evaluate the ECSA of the

catalyst, Cu-UPD was performed on the Ru-Co$_3$C@CNT and Ru-Cr$_{23}$C$_6$@CNT catalysts, respectively, (Supplementary Fig. 32a-b). By calculation, the ECSA of Ru-Co$_3$C@CNT and Ru-Cr$_{23}$C$_6$@CNT catalysts were 76.2 and 88.1 m$^2$g$^{-1}$$_{Ru}$, respectively, (Supplementary Fig. 32c). The TOF value was obtained by the same method, as shown in Supplementary Fig. 30d, Ru-Co$_3$C@CNT, Ru-Cr$_{23}$C$_6$@CNT, and Ru-Mo$_2$C@CNT all showed a large TOF value, which was 10.3, 9.2, and 21.9 s$^{-1}$ under the overpotential of 100 mV, respectively. This result proved that these materials all had high intrinsic catalytic activity. In addition, double-layer capacitance (C$_{dl}$) was used to further evaluate the active surface area of the material. In 1.0 M KOH solution, the capacitors (C$_{dl}$) of Ru-Co$_3$C@CNT, Ru-Cr$_{23}$C$_6$@CNT, and Ru-Mo$_2$C@CNT were 14.5, 10.8, and 17.2 mF cm$^{-2}$, respectively, (Supplementary Fig. 33), indicating that these three samples all had a large active specific surface area. Besides, the electrochemical impedance spectroscopy (EIS) (Supplementary Fig. 34) showed that both Ru-Co$_3$C@CNT (24.1 Ω) and Ru-Cr$_{23}$C$_6$@CNT (22.4 Ω) also had smaller impedance values, indicating that such materials had higher charge transfer rates and easier HER reaction kinetics. Unexpectedly, the three catalysts prepared by microwave pyrolysis have high electrochemical stability. As shown in Supplementary Fig. 30e, after 10,000 cycles of scanning, the polarization curves of Ru-Co$_3$C@CNT and Ru-Cr$_{23}$C$_6$@CNT were almost not shifted. The current-time (i–t) test (Supplementary Fig. 30f) showed that the current density remained almost constant for 100 h. All the results show that this kind of catalyst has high long-term stability.

**Large-scale production of Ru-Mo$_2$C@CNT catalyst for HER**. Finally, in order to prove that the Ru-Mo$_2$C@CNT catalyst can be produced on a large scale, the raw material was expanded by 200 times and the experiment was carried out to obtain about 1.6044 g of product (Fig. 4a), with a yield of up to 80.2%. The production method of this catalyst is fast, solvent-free and efficient, has low requirements on experimental equipment, and can realize high-throughput production, which greatly reduces its production cost. In addition, in order to further explore the Ru-Mo$_2$C@CNT catalyst can be used for large-scale industrial production of H$_2$, the catalyst was directly drip-coated on the foamed nickel (NF) (Fig. 4b), which HER performance was tested in 1.0 M KOH electrolyte (Fig. 4c). When the Ru-Mo$_2$C loading amount in the catalyst supported on NF is 0.95 mg cm$^{-2}$, the catalyst Ru-Mo$_2$C@CNT achieved the industrial current densities of 500 mA cm$^{-2}$ and 1000 mA cm$^{-2}$ at low overpotentials of 56 mV and 78 mV (Fig. 4d), respectively, which were better than all the catalysts reported (Fig. 4h and Supplementary Table 5). Moreover, gas chromatography technology was used to evaluate the gas products and the corresponding Faraday efficiency in 1.0 M KOH electrolyte (Fig. 4e). It can be reflected from this figure that the efficiency of electron conversion involved in the catalytic reaction is determined to be close to 100%, which means that almost all electrons are used to generate hydrogen in the water electrolysis process.

Furthermore, in order to study the stability of the catalyst Ru-Mo$_2$C@CNT, a volt-ampere characteristic curve scan was performed after 10,000 cycles of CV cycles and chronopotentiometric test and chronoamperometric test were performed. After 10,000 cycles of CV, the LSV curve basically did not shift, which further proved the stability of the Ru-Mo$_2$C@CNT electrocatalyst (Fig. 4f). More importantly, the activity can be well maintained for more than 1000 h without decay at the large current density of 500 mA cm$^{-2}$ (Fig. 4g). In addition, as shown in Supplementary Fig. 35, the chronopotentiometric curve of the Ru-Mo$_2$C@CNT electrode was tested at a constant current density of 500 mA cm$^{-2}$

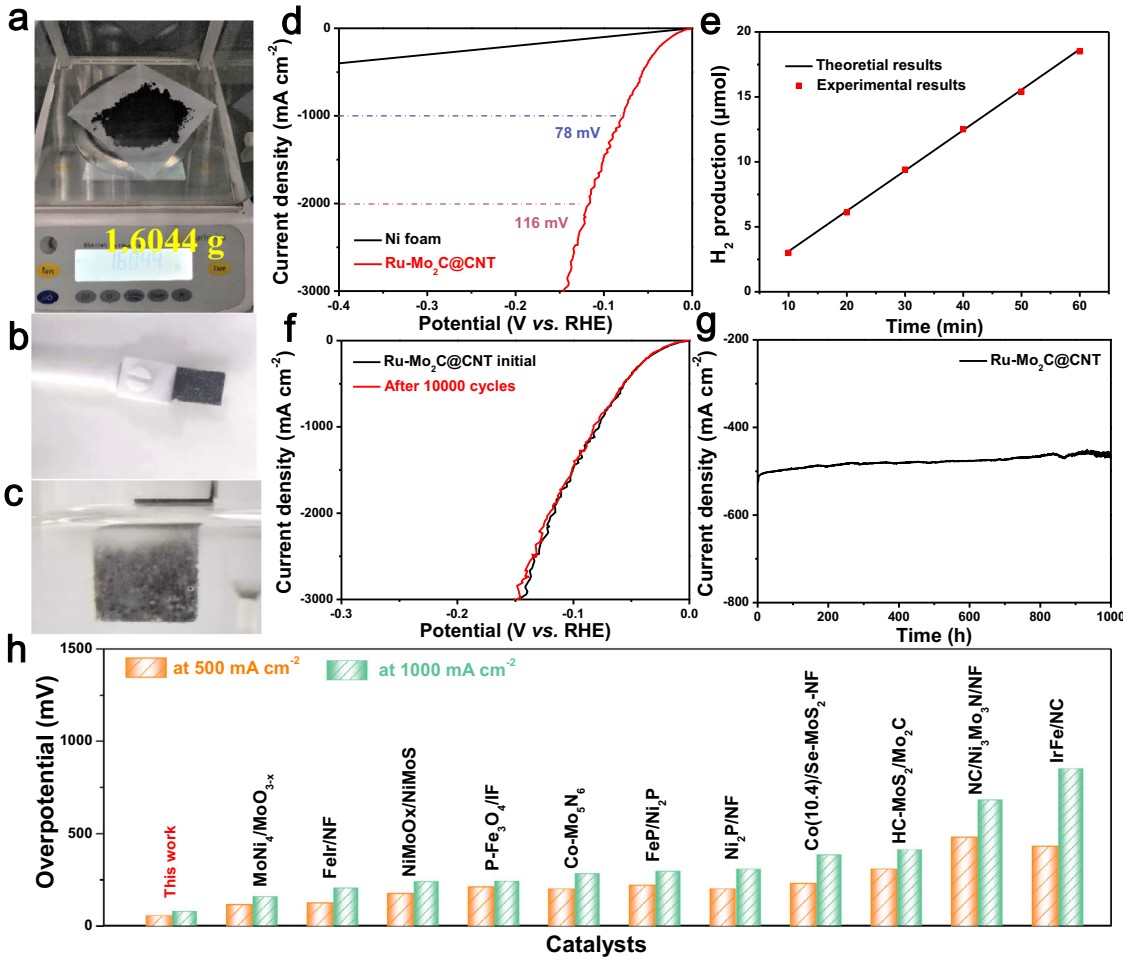

**Fig. 4 Test of large-scale production of Ru-Mo₂C@CNT catalyst for HER. a** Photograph of catalyst output. **b** Photo of Ru-Mo₂C@CNT catalyst directly drip-coated on the NF. **c** Photo of H₂ bubbles leaving the surface of Ru-Mo₂C@CNT catalyst. **d** Polarization curves for Ru-Mo₂C@CNT in 1.0 M KOH solution. **e** Theoretical and experimental results of H₂ production of Ru-Mo₂C@CNT. **f** Polarization curves for Ru-Mo₂C@CNT before and after 10,000 cycles. **g** The current-time (i–t) curve of Ru-Mo₂C@CNT under the temporal evolution of the potential required to maintain 500 mA cm⁻² for 1000 h. **h** In 1.0 M KOH solution, comparison of the overpotentials at 500 mA cm⁻² and 1000 mA cm⁻² for the recent reported catalysts.

for 500 h. And this result further prove that the Ru-Mo₂C@CNT material has high stability. Moveover, as shown in Supplementary Fig. 36, the comparative samples Ru@CNT and Mo₂C@CNT also showed high electrochemical stability. In addition, the SEM and TEM images of Ru-Mo₂C@CNT after the long-term stability test showed no change in the morphology of the material, and the XRD pattern and XPS spectra further reflected that the structure of this material did not change (Supplementary Figs. 37-38). This discovery shows that the catalyst synthesized by microwave pyrolysis has a high stability, and provide a feasible way for the design of other highly stable materials.

## Discussion

In summary, an ultra-small (3.5 nm) Ru-MₓC@CNT (M = Mo, Co, Cr) catalyst with heterogeneous structure and strong metal-support interaction was synthesized by a simple, fast, environmentally friendly solid-phase microwave pyrolysis method. A series of physical characterization and chemical tests show that the unique heterojunction synergy and the SMSI between the nano-heterojunction particles and the CNT substrate of Ru-Mo₂C@CNT composite materials lead to this material has superior electronic conductivity, abundant active sites and stable structure, which are the fundamental reasons for its high

electrocatalytic HER activity and stability. The catalyst Ru-Mo₂C@CNT only showed an overpotential of 15 mV at a current density of 10 mA cm⁻², and its TOF value is as high as 21.9 s⁻¹ at an overpotential of 100 mV in 1.0 M KOH alkaline solution. In addition, this catalyst can reach high current densities of 500 and 1000 mA cm⁻² at low overpotentials of 56 and 78 mV, respectively. Moreover, it has a high stability which the current density remains basically unchanged after the 1000 h i–t test. Ultra-small heterojunction nanocatalysts with SMSI effect are synthesized by a simple, fast, and solvent-free microwave pyrolysis method, which provides a feasible way for the reasonable design of other large-scale production catalysts.

## Methods

**Materials**. Molybdenumhexacarbonyl (Mo(CO)₆, 98%) was bought from Sigma–Aldrich. Hexacarbonylchromium (Cr(CO)₆, 99%) and Octacarbonyldicobalt (Co₂(CO)₈) were supplied by Alfa Aesar. Triruthenium dodecacarbonyl (Ru₃(CO)₁₂, 98%) was purchased from Aladdin. Carbon nanotube was bought from Aladdin.

**Preparation of MWCNT**. Disperse 50 mg MWCNT powder in a mixed solution with a concentration of H₂SO₄: HNO₃ = 3:1 for ultrasonic treatment for 1–2 h, and then expose it to 1 M HCl for ultrasonic treatment for 30 min. Finally, filter the acidified CNT, wash it with ionized water until pH = 7, and dry it at 60 °C for 12 h.

**Preparation of Ru-Mo$_2$C@CNT**. First, 10 mg of processed MWCNT, 5 mg Mo(CO)$_6$, and 5 mg Ru$_3$(CO)$_{12}$ were mixed and ground in a mortar for 30 min to mix evenly. Then, the mixture was put into a 10 mL quartz bottle and microwaved in a household microwave oven for 100 s.

**Preparation of Ru@CNT and Mo$_2$C@CNT**. First, mix 10 mg of processed MWCNT and 5 mg Ru$_3$(CO)$_{12}$ or 5 mg Mo(CO)$_6$, grind it in a mortar for 30 min and mix evenly, then put it in a 10 mL quartz bottle and microwave it in a household microwave oven 100 s.

**Preparation of Ru-Mo$_2$C@CNT in different CNT proportions**. Respectively using 5 mg of Ru$_3$(CO)$_{12}$, 5 mg of Mo(CO)$_6$, and 5 mg/10 mg/20 mg MWCNT mixing, grinding in the mortar homogeneously 30 min, and then puts it into 10 mL of quartz in the bottle, and microwave processing in the microwave oven for 100 s.

**Preparation of Ru-Mo$_2$C@CNT in different Ru:Mo$_2$C proportions**. Respectively using 4 mg of Ru$_3$(CO)$_{12}$ and 8 mg of Mo(CO)$_6$, 6 mg of Ru$_3$(CO)$_{12}$ and 6 mg Mo(CO)$_6$ and 8 mg of Ru$_3$(CO)$_{12}$ and 4 mg Mo(CO)$_6$ and 12 mg MWCNT mixing, grinding in the mortar homogeneously 30 min, and then puts it into 10 mL of quartz in the bottle, and microwave processing in the microwave oven for 100 s.

**Preparation of Ru-Co$_3$C@CNT and Ru-Cr$_{23}$C$_6$@CNT**. Mo(CO)$_6$ in the preparation process of Ru-Mo$_2$C@CNT catalyst was directly replaced with Co$_2$(CO)$_8$ or Cr(CO)$_6$, and the remaining steps were synthesized in the same operation method.

**Characterization**. To study the morphology and structure of the catalyst, a scanning electron microscope (SEM) was tested on Hitachi S-4800 instrument. The transmission electron microscope (TEM) and high-resolution TEM (HRTEM) of the catalyst were tested using FEI Tecnai-G2 F30 at an accelerating voltage of 300 KV, and the structure of the catalyst was further characterized. Powder X-ray diffraction (XRD) pattern recording was performed on an X'Pert-PRO MPD diffractometer, which was run with Cu Kα radiation at 40 KV and 40 mA. The content of elements is determined by inductively coupled plasma atomic emission spectrometer (ICP-AES, Varian 710-ES). X-ray photoelectron spectroscopy (XPS) analysis was performed with an Axis Supra spectrometer using a monochromatic Al Kα source at 15 mA and 14 kV. Scan analysis with an analysis area of 300 × 700 microns and a pass energy of 100 eV. The spectrum was calibrated by carbon 1 s spectrum, and its main line was set to 284.8 eV, and then the valence state of the catalyst was analyzed using Casa XPS software. The catalyst that has been tested for stability is scraped from the working electrode by ultrasonic treatment and collected for the next step of SEM, TEM, and XRD characterization.

**Electrochemical measurements**. Disperse 1 mg of the catalyst in 1 mL of a mixed solution of isopropanol + ultrapure water + 5% Nafion (v:v:v = 3:1:0.05), after sonication for 1 h, the different catalysts with the concentration of 1 mg mL$^{-1}$ was obtained.

Electrochemical measurements were carried out in a conventional three-electrode battery of a CHI 760E Electrochemical Workstation (Shanghai Chenhua Instrument Corporation, China). A graphite rod electrode was used as the counter electrode, and the reference electrode was a saturated calomel electrode (SCE). A glassy carbon electrode (GCE, diameter: 3 mm, area: 0.07065 cm$^2$) was used as the working electrode. Take 10 μL of the mixed slurry and drop it evenly on the surface of the GCE. After it is naturally dried, further electrochemical tests are performed. All potentials reported in this work are corrected using reversible hydrogen electrodes (RHE). In a 1.0 M KOH solution saturated with N$_2$, linear sweep voltammetry (LSV) was used to test and evaluate the HER performance of the catalyst at a sweep rate of 5 mV s$^{-1}$. All polarization curves were corrected for 95% iR. The durability test was performed in 1.0 M KOH solution using chronoamperometry. In addition, the LSV after 10,000 cycles of CV was measured to further evaluate the stability of the catalyst. Electrochemical impedance spectroscopy (EIS) measurement was performed at a frequency of 0.1 Hz to 100 kHz in a 1.0 M KOH solution saturated with N$_2$.

**Electrochemical hydrogen production/oxidation (HER/HOR) reaction test**. At this time, a disk electrode (RDE area: 0.196 cm$^2$) was used as the working electrode. In 1.0 M KOH electrolyte saturated with N$_2$, the CV scan was performed at a scan rate of 100 mV/s from 0.05 V to 1.10 V vs. RHE until it stabilized. The HER/HOR test was performed by linear sweep voltammetry (LSV) in a 1.0 M KOH solution saturated with H$_2$ using a sweep rate of 10 mV/s, a rotation speed of 1600 rpm, and all data were iR corrected. The exchange current density (J$_0$) and the symmetry factor (α) are obtained by fitting kinetic current density (j$_k$) at small current density region into the Butler-Volmer equation as follows:

$$j_k = j_0(e^{\frac{\alpha F\eta}{RT}} - e^{\frac{(\alpha-1)F\eta}{RT}}) \tag{1}$$

The current density (j) obtained from the working electrode is the sum of two currents: dynamic current density (j$_k$) and diffusion current density (j$_d$). The dynamic current density (j$_k$) is derived from the following Koutecky–Levich

equation:

$$\frac{1}{j} = \frac{1}{j_k} + \frac{1}{j_d} \tag{2}$$

in which j$_d$ obeys the Levich equation:

$$j_d = 0.62n\text{FD}^{2/3}v^{-1/6}C_0\omega^{1/2} \tag{3}$$

in which F is the Faraday constant (96,485 C mol$^{-1}$), n is the number of electrons involved in the oxidation reaction, C$_0$ is the H$_2$ concentration in solution, D is the diffusion coefficient of the reactant (cm$^2$s$^{-1}$), v is the viscosity of the electrolyte (cm$^2$s$^{-1}$), and ω is the rotation speed (rpm).

**Active sites calculations**. The underpotential deposition (UPD) of copper (Cu) was used to calculate the active sites of the Ru-Mo$_2$C@CNT and other comparative samples. In this method, the number of active sites (n) can be calculated based on the UPD copper stripping charge (Q$_{Cu}$, Cu$_{upd} \rightarrow$ Cu$^{2+}$+2e$^-$) using the following equation.

$$n = \frac{Q_{Cu}}{2*F} \tag{4}$$

where F is the Faraday constant (96,485.3 C mol$^{-1}$).

**ECSA measurements**. The ECSA of the catalyst has been already proved could be calculated by Cu underpotential deposition (UPD) method. The ECSAs of catalysts were calculated by integrating the charge associated with oxidation of Cu (on the surface of catalyst by Cu-UPD) in electrolyte containing 50 mM CuSO$_4$ and 0.5 M H$_2$SO$_4$, by assuming a charge of 420 μC cm$^{-2}$. The ECSA can be calibrated as:

$$\text{ECSA(cm}^2_{\text{metal}}/\text{g}_{\text{metal}}) = \frac{Q_{Cu}}{M_{\text{metal}} \times 420\,\mu C\,\text{cm}^{-2}} \tag{5}$$

where M$_{metal}$ is the mass loading of metal on a certain geometric area of the working electrode.

**Measurement of the turnover frequency (TOF)**. The TOF (s$^{-1}$) was calculated by the following formula.

$$\text{TOF(s}^{-1}) = \frac{I}{2*F*n} \tag{6}$$

where I is the current (A) during linear sweep voltammetry (LSV), F is the Faraday constant (96,485.3 C mol$^{-1}$), n is the number of active sites (mol). The factor 1/2 is based on the assumption that two electrons are necessary to form on hydrogen molecules.

To obtain monolayer of copper, Ru-Mo$_2$C@CNT was first polarized at 0.23 V for 100 s. For the given polarization potential, there were only two oxidation peaks related to bulk and monolayer of Cu.

## Data availability
The data that support the findings of this study are available from the corresponding author upon reasonable request.

## Code availability
All code supporting the findings of this study are available from the corresponding author on request.

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

## Acknowledgements

This work was supported by the National Natural Science Foundation of China (22001143, 21571112, 51572136, 51772162, 51802171), the Taishan Scholars Program, Natural Science Foundation of Shandong Province, China (ZR2018BB031, ZR2019JQ14), Open Fund of the Key Laboratory of Eco-chemical Engineering (Qingdao University of Science and Technology, No. KF1702), the Taishan Scholar Project of Shandong Province (tsqn201909123).

## Author contributions

L.W. and J.L. supervised the research. J.L. conceived the research. J.L. and X.W. designed the experiments. X.W. performed most of the experiments and data analysis. Z.W., D.Z., and Y.Q. prepared the electrodes and helped with electrochemical measurements. M.W., Y.H., and T.Z. helped analyze physical characterization data. B.Y. and S.L. helped answer some questions. All authors discussed the results and commented on the manuscript.

## Competing interests

The authors declare no competing interests.
