## [Peer Review File · Nature Communications]

REVIEWER COMMENTS

Reviewer #1 (Remarks to the Author):

For authors

The manuscript "Solvent-free microwave synthesis of ultra-small Ru-Mo₂C@CNT with strong metal-support interaction for industrial alkaline hydrogen evolution reaction" describes the fabrication of Ru-Mo alloy nanoparticles attached to carbon nanotubes through the decomposition of Mo and Ru of carbonyls in a household microwave oven. The group characterizes these nanoparticle-decorated CNTs to show the bonding and incorporation of the nanoparticles. Electrochemical analysis in strong basic solutions showed superior hydrogen generation compared to catalysts from similar research. The manuscript was easy to follow and gave a clear picture of the aims and results.

This manuscript is another step in a long line of studies on alloy nanoparticles bound to CNTs for catalysis purposes. That being said, the chemical composition of the catalyst is novel, and the electrochemical activity of these structures show notable improvement compared to past publications. Therefore, I recommend publishing this manuscript once the authors have considered these comments and made the corresponding minor modifications.

-Many of the figures show a detailed study of the catalyst that brings great insight to its properties and how it fares compared to catalysts from similar publication. However, there is a large amount of overlap in the data shown in many of the graphs. These redundant representations might be distracting to the reader. I recommend moving some of these figures to the supplementary information section of this manuscript.

-Fabrication of nanoparticles without the use of organic solvents has been referred to as green chemistry in many scientific papers, but there should be stricter criteria when considering what reaction can be called green. In the case of this paper, the authors are advised to retrace the steps back and consider the environmental impacts of precursor synthesis and confirm whether it is green.

-The use of nafion in the fabrication of the working electrode, although a common method of binding nanoparticles to substrates, is not applicable in industrial settings and may negate the target set by the authors as a cheap and straightforward process of catalyst fabrication.

-The XPS data representation doesn't seem to be consistent. In plots a, b, and d of figure 2, the raw data is represented in black, whereas in plot c, it is red. Using consistent representation in graphs is highly advised. The XPS signal to noise ratio for the CNT is much higher than the other samples, and it seems that the same characterization conditions were not used for all samples. I recommend the authors to revisit the data and confirm its soundness.

-EDX shows a wide view of a CNT decorated with nanoparticles. To better show the presence of the elements in question in these particles, it is highly recommended to include a closeup image of one or few nanoparticles with overlapping element colormaps. This would provide a more effective visual representation of the distribution.

Reviewer #2 (Remarks to the Author):

The manuscript reported by Wu et al. addresses a very important research topic for hydrogen economy. However, I really doubt that the presented manuscript is not suitable for the broad readership in Nature Communications. Although the authors have prepared a new material

(RuMo₂C@CNT), its catalytic properties are not sufficient to meet and/or to exceed the state-of-the-art electrocatalysts towards hydrogen evolution reaction (HER) in alkaline environment. This observation is very likely related to the lack of knowledge in the field of electrochemistry. The referee is very surprised about the extremely poor HER performance of pure Pt/C which is actually well-established and frequently reported (e.g. see Nature Communications volume 11, Article number: 1278 (2020) for Pt/C data). In addition, the authors have analyzed their electrochemical data with a minimum effort. Excellent and very good electrochemical data for HER/HOR are typically shown including Butler-Volmer plots, exchange current density with normalization of metal mass and number of catalytically active sites as well as charge transfer coefficient to get a deeper insight into the kinetics and mechanism of HER. Furthermore, the determination of the ECSA via underpotential deposition of Cu is completely wrong. In Figure S16 and S28, the anodic peaks of a Cu monolayer appear at the same potentials like the redox potential of Cu/Cu²⁺. The authors mainly observed a bulk dissolution of metallic Cu and this fact explains the high ECSA values. In Figure S32, the Nyquist plot starts for all materials nearly at Z' = 0 Ohm. This is an observation which can not be explained by the presented data. It is also frustrated to see that the authors did not make enough effort to measure their materials in a proper manner. The Raman, XRD, EDX and other data are presented in a poor quality.

Reviewer #3 (Remarks to the Author):

Electrocatalytic hydrogen evolution reaction (HER) by splitting water has become an effective method for the sustainable production of H₂. It is highly desirable and imperative to develop new HER electrocatalysts with low-cost and high-performance. Herein, the authors reported a simple, fast and solvent-free microwave pyrolysis method for the synthesis of ultra-small (3.5 nm) Ru-M_xC@CNT (M=Mo, Co, Cr) catalyst with heterogeneous structure and strong metal-support interaction in one step. The fabricated Ru-Mo₂C@CNT catalyst exhibits a low overpotential of 15 mV at a current density of 10 mA cm⁻², and exhibits a large TOF value up to 57.8 s⁻¹ under an overpotential of 100 mV. This paper is interesting, and I recommend this paper can be accepted after following revisions:

1. The author claims that the synthesis of Ru-Mo₂C@CNT, however, in Figure 1, only one Ru-Mo₂C nanoparticle is seen in Figure 1c, indicating that the Ru-Mo₂C nanoparticles are not ubiquitous on CNT. So the TEM image with more Ru-Mo₂C nanoparticles should be provided. This is crucial for this paper.
2. For Ru-Mo₂C@CNT, how about the content of Ru or Mo₂C on the catalytic activity and stability for HER? The authors should provide more information.
3. How about the long-term durability of Ru-Mo₂C@CNT for HER in alkaline media? As shown in Figure 4f, the test time of durability is too short, and the authors should measure the durability of HER for at least 50 hours.
4. How do you prove the structure stability of Ru-Mo₂C@CNT for long-term durability of HER? Please provide some evidences for structure stability of Ru-Mo₂C@CNT after durability test.
5. The loading of Ru-Mo₂C on CNTs should be provided for Figure 4, and the effect of the loading of Ru-Mo₂C on CNTs on the catalytic activity and stability should be studied.
6. For the data shown in Figure 5g, to maintain the current density at 500 mA cm⁻², the overpotential should be provided.
7. Why does Ru-Mo₂C have higher catalytic performance than the other Ru-Co₃C@CNT, Ru-Cr₂₃C₆@CNT? Please provide some explains in the paper.
8. Some relevant references about hydrogen evolution electrocatalysis may be considered to be cited, such as Angew. Chem. Int. Ed. 2017, 56, 2960; Angew. Chem. Int. Ed. 2017, 56, 8120; J. Am. Chem. Soc. 2018, 140, 5118.
9. For supplementary Figure 21, the TEM image of Ru-Co₃C@CNT does not show the existence of heterojunction, please provide another TEM.

Lei Wang

College of Chemistry and Molecular Engineering

Qingdao University of Science and Technology, Qingdao 266042, P. R. China.

E-mail: inorchemwl@126.com

Apr. 6, 2021

Dear Editor,

We highly appreciate your kind consideration and review on our paper entitled “*Solvent-free microwave synthesis of ultra-small Ru-Mo₂C@CNT with strong metal-support interaction for industrial alkaline hydrogen evolution reaction*” (NCOMMS-20-48819). We have carefully revised the manuscript and responded all the raised valuable comments by three reviewers. We have also highlighted the changes in manuscript and supporting information with red color. Thanks to the valuable suggestions, our manuscript could be significantly improved. We response to the reviewers’ comments point by point and highlight the changes in the revised manuscript. The list of changes and our responses to three reviewers’ comments are provided as follows.

Reply to Reviewers' Comments

Dear Reviewers,

Thank you for your precious time to constructive comments on our manuscript titled “**Solvent-free microwave synthesis of ultra-small Ru-Mo₂C@CNT with strong metal-support interaction for industrial alkaline hydrogen evolution reaction**” (Manuscript ID: NCOMMS-20-48819) for *Nature Communications*. We sincerely appreciate your opinions and confirmation of our work. Accordingly, we have supplied the corresponding response and revision based on the comments. We sincerely hope that our responses will fully address your concerns about our work.

To Reviewer 1:

General Comment: The manuscript "Solvent-free microwave synthesis of ultra-small Ru-Mo₂C@CNT with strong metal-support interaction for industrial alkaline hydrogen evolution reaction" describes the fabrication of Ru-Mo alloy nanoparticles attached to carbon nanotubes through the decomposition of Mo and Ru of carbonyls in a household microwave oven. The group characterizes these nanoparticle-decorated CNTs to show the bonding and incorporation of the nanoparticles. Electrochemical analysis in strong basic solutions showed superior hydrogen generation compared to catalysts from similar research. The manuscript was easy to follow and gave a clear picture of the aims and results.

This manuscript is another step in a long line of studies on alloy nanoparticles bound to CNTs for catalysis purposes. That being said, the chemical composition of the catalyst is novel, and the electrochemical activity of these structures show notable improvement compared to past publications. Therefore, I recommend publishing this manuscript once the authors have considered these comments and made the corresponding minor modifications.

Author Reply: Thanks for your positive comments and support for our work. To further improve the quality of this manuscript as well as address your concerns, we have revised our manuscript as your suggestions. In addition, we also supplied a point-by-point response as follows. We wished the revised manuscript can fulfill your high requirements for the publication in *Nature Communications*.

Question 1: Many of the figures show a detailed study of the catalyst that brings great insight to its properties and how it fairs compared to catalysts from similar publication. However, there is a large amount of overlap in the data shown in many of the graphs. These redundant representations might be distracting to the reader. I recommend moving some of these figures to the supplementary information section of this manuscript.

Author Reply 1: Thanks very much for reviewer's helpful comment and suggestion. In order to allow readers to read the article more clearly, we have moved **Figure 4** to **Supplementary Figure 28** in the supporting information.

Question 2: Fabrication of nanoparticles without the use of organic solvents has been referred to as green chemistry in many scientific papers, but there should be stricter criteria when considering what reaction can be called green. In the case of this paper, the authors are advised to retrace the steps back and consider the environmental impacts of precursor synthesis and confirm whether it is green.

Author Reply 2: Thanks for reviewer's valuable comment and suggestion. Considering that organic solvents are used in the synthesis of precursors, the term green in the article is indeed not appropriate. Without considering the precursor synthesis, we have changed the green characteristics of the catalyst synthesis method in the article to solvent-free characteristics.

Manuscript Revision: The corresponding sentence "Exploring a simple, fast, solvent-free synthetic method for large-scale preparation of cheap, highly active electrocatalysts for industrial hydrogen evolution reaction (HER) is one of the most promising work in the field of electrocatalysis today." has been corrected in the revised manuscript (please see **15-17 lines** (the red-label part) of **Page 1** in the revised manuscript), The corresponding sentence "This whole synthesis process only requires a simple step, no solvent is involved, and it is very fast, solvent-free and high yield." has been corrected in the revised manuscript (please see **1 line** (the red-label part) of **Page 5** in the revised manuscript) and "The production method of this catalyst is fast, solvent-free and efficient, has low requirements on experimental equipment, and can realize high-throughput production, which greatly reduces its production cost." has been corrected in the revised manuscript (please see **12-14 lines** (the red-label part) of **Page 12** in the revised manuscript).

Question 3: The use of nafion in the fabrication of the working electrode, although a common method of binding nanoparticles to substrates, is not applicable in industrial settings and may negate the target set by the authors as a cheap and straightforward process of catalyst fabrication.

Author Reply 3: Thanks for reviewer's valuable comment and suggestion. In consideration of the reviewer's significant question about the use of nafion, our next goal is to directly grow the catalyst on a macroscopic three-dimensional support (such as carbon cloth or nickel foam and other inexpensive supports) during the synthesis process.

Question 4: The XPS data representation doesn't seem to be consistent. In plots a, b, and d of figure 2, the

raw data is represented in black, whereas in plot c, it is red. Using consistent representation in graphs is highly advised. The XPS signal to noise ratio for the CNT is much higher than the other samples, and it seems that the same characterization conditions were not used for all samples. I recommend the authors to revisit the data and confirm its soundness.

Author Reply 4: Thank you for reviewer's helpful suggestion. The retested data and the corrected plot c have been shown in **Figure R1**. In addition, we re-characterized the XPS data of CNTs (**Figure R1a**) using the same test conditions as the other catalysts in the article.

Figure R1. High-resolution XPS spectra of Ru-Mo₂C@CNT. (a) High resolution XPS spectra of C 1s of CNT and Ru-Mo₂C@CNT. (b) High resolution XPS spectra of Ru 3d spectra of Ru@CNT and Ru-Mo₂C@CNT. (c) High resolution XPS spectra of Ru 3p of Ru@CNT and Ru-Mo₂C@CNT. (d) High resolution XPS spectra of Mo 3d of Mo₂C@CNT and Ru-Mo₂C@CNT.

Manuscript Revision: The **Figure R1** has been corrected in the Manuscript (please see **Figure 2**).

Question 5: EDX shows a wide view of a CNT decorated with nanoparticles. To better show the presence of the elements in question in these particles, it is highly recommended to include a closeup image of one or few nanoparticles with overlapping element colormaps. This would provide a more effective visual representation of the distribution.

Author Reply 5: Thanks for your kind comments and suggestion. As shown in Figure R2, in order to better show the distribution of each element in the catalyst, we re-tested the EDX mapping of Ru-Mo₂C@CNT, and provided the element color map of the simultaneous presence of Ru and Mo elements.

Figure R2. HAADF-STEM image and corresponding EDX maps of Ru-Mo₂C@CNT for Ru, Mo and Ru+Mo, respectively.

Manuscript Revision: The **Figure R2** has been corrected in the Manuscript (please see **Figure 1e.**).

To Reviewer 2:

General Comment: The manuscript reported by Wu et al. addresses a very important research topic for hydrogen economy. However, I really doubt that the presented manuscript is not suitable for the broad readership in Nature Communications.

Author Reply: Many thanks for taking time to access our manuscript and providing thoughtful feedbacks. The reviewer's comments were helpful to improve the quality of our manuscript. Based on your valuable comments, we have supplied the detailed.

Question 1: Although the authors have prepared a new material (Ru-Mo₂C@CNT), its catalytic properties are not sufficient to meet and/or to exceed the state-of-the-art electrocatalysts towards hydrogen evolution reaction (HER) in alkaline environment. This observation is very likely related to the lack of knowledge in the field of electrochemistry.

Author Reply 1: Thank you for your helpful comment. This new catalyst (Ru-Mo₂C@CNT) we designed and synthesized (the overpotential only is 15 mV at 10 mA cm⁻²) becomes among the best HER electrocatalysts in alkaline solution (**Supplementary Table 4**). More importantly, when the catalyst Ru-Mo₂C@CNT was loaded on the Ni foam, the overpotential exhibited at large currents of 500 mA cm⁻² and 1000 mA cm⁻² is the lowest in the reported literature (**Figure 4h**). In terms of the simplicity of the synthesis method, the environmental friendliness of the process, the high yield and the high catalytic activity for HER, the research of Ru-Mo₂C@CNT is of great significance in practical applications.

Question 2: The referee is very surprised about the extremely poor HER performance of pure Pt/C which is actually well-established and frequently reported (e.g. see Nature Communications volume 11, Article number: 1278 (2020) for Pt/C data).

Author Reply 2: Thanks for your valuable comment. We repurchased a new commercial Pt/C from Aladdin. And we re-provide the LSV curve (**Figure R1a**), overpotential at 10 mA cm⁻² (**Figure R1c**) and 20 mA cm⁻² (**Figure R2**), exchange current density (**Figure R1c**), Tafel slope (**Figure R1b**) and TOF (**Figure R1e**) data of commercial Pt/C. Among them, the LSV curve of commercial Pt/C shows its better electrochemical HER activity. The overpotential of commercial Pt/C at 10 mA cm⁻² and 20 mA cm⁻² are 33 mV and 60 mV, respectively, and the Tafel slope is 44 mV dec⁻¹, which are similar to the performance in the literature (Nat. Commun. 2020, 11, 1278). In addition, the exchange current density of commercial Pt/C is 2.23 mA cm⁻², and the TOF value at 100 mV shows 8.45 s⁻¹. The comparison with the properties of the catalyst synthesized in this article is shown in the figure below.

Figure R1. Electrocatalytic HER performance test of catalysts in N_2 -saturated 1.0 M KOH solution. (a) HER polarization curves of CNT, $Mo_2C@CNT$, $Ru@CNT$, Pt/C and $Ru-Mo_2C@CNT$ catalysts. (b) Comparison of overpotential changes at 10 mA cm^{-2} and exchange current density. (c) Tafel plots obtained from the polarization curves in (a). (d) In 1.0 M KOH solution, compared with the recently reported HER catalyst Tafel slope and overpotential at 10 mA cm^{-2} . (e) The potential dependent TOF curves of $Mo_2C@CNT$, $Ru@CNT$, Pt/C and $Ru-Mo_2C@CNT$. (f) Polarization curves for $Mo_2C@CNT$, $Ru@CNT$, Pt/C and $Ru-Mo_2C@CNT$ catalysts before and after 10000 cycles.

Figure R2. Comparison of overpotential changes at 20 mA cm⁻² of Ru-Mo₂C@CNT, Pt/C, Ru@CNT and Mo₂C@CNT catalysts.

Manuscript Revision: We changed the original sentences to “Correspondingly, Ru-Mo₂C@CNT catalyst showed lower overpotential than commercial Pt/C (33 mV), which further indicated that Ru-Mo₂C@CNT had an excellent electrocatalytic HER activity.” and “In 1.0 M KOH solution, Ru-Mo₂C@CNT only showed a low Tafel slope of 26 mV dec⁻¹ (**Figure 3b**), which was much lower than Pt/C (44 mV dec⁻¹), Ru@CNT (76 mV dec⁻¹) and Mo₂C@CNT (197 mV dec⁻¹), which implies that the Tafel-volmer mechanism is present in the catalytic process, and the electrochemical desorption of H₂ is the rate determining step.” in the revised manuscript (please see **2-3 lines** and **7-9 lines** (the red-label part) of **Page 9** in the revised manuscript).

The **Figure R1** and **Figure R2** have been corrected in the Manuscript and the Supporting Information, respectively. (please see **Figure 2** and **Supplementary Figure 16**.)

Question 3: In addition, the authors have analyzed their electrochemical data with a minimum effort. Excellent and very good electrochemical data for HER/HOR are typically shown including Butler-Volmer plots, exchange current density with normalization of metal mass and number of catalytically active sites as well as charge transfer coefficient to get a deeper insight into the kinetics and mechanism of HER.

Author Reply 3: Thanks for reviewer’s valuable comment and suggestion. We obtained the exchange current density of the catalyst by extrapolating the Tafel diagram. As shown in **Figure R1c** and **R3**, the exchange current density has been normalized by the electrochemical surface area of the supported nanocatalyst of this series of materials, and the result shows that the exchange current density of Ru-Mo₂C@CNT is 4.4 mA cm⁻², which is greater than other catalysts in this article. In order to compare the exchange current density of materials in other literatures more clearly, the exchange current density of the catalyst Ru-M_xC@CNT (M=Mo,Co,Cr) and other materials in the literatures are listed in the **Table R1**. In addition, we calculated the exchange current density (*j*₀) with normalization of catalyst mass of the

catalyst Ru-Mo₂C@CNT in this paper is 440 mA cm⁻² mg⁻¹, which is also greater than the commercialized Pt/C (223 mA cm⁻² mg⁻¹).

Moreover, the number of active sites (n) of the catalyst is calculated by the following formula. $n = \frac{Q_{Cu}}{2 \cdot F}$. As shown in **Figure R4**, the catalyst Ru-Mo₂C@CNT has the largest number of active sites, which is 2.1*10⁻³ mol g⁻¹_{Ru}, which is larger than commercial Pt/C (1.6*10⁻³ mol g⁻¹_{Pt}). The number of active sites for other catalysts are shown in **Figures R4** and **R5**.

Due to the limitation of experimental conditions, we cannot carry out the HOR reaction of the material, nor can we obtain a complete Butler-Volmer plots and charge transfer coefficient. However, from the Tafel slope and the above exchange current density and the number of active sites of the catalysts, we deeply explained the reaction kinetics and mechanism of HER from a certain aspect, which also shows that the catalyst has excellent electrocatalytic HER activity.

Figure R3. Comparison of overpotential changes at 10 mA cm⁻² and exchange current density of Ru-Co₃C@CNT, Ru-Cr₂₃C₆@CNT and Ru-Mo₂C@CNT catalysts.

Figure R4. The active sites using the Cu-UPD method of Ru-Mo₂C@CNT, Pt/C, Ru@CNT and Mo₂C@CNT.

Figure R5. The active sites using the Cu-UPD method of Ru-Mo₂C@CNT, Ru-Cr₂₃C₆@CNT and Ru-Co₃C@CNT.

Table R1. The exchange current densities of various samples.

Catalyst	j_0 , normalized (mA cm^{-2})	Reference
Ru-Mo ₂ C@CNT	4.4	This work
Ru-Cr ₂₃ C ₆ @CNT	3.8	This work
Pt/C	2.2	This work
Ru@CNT	2.0	This work
Mo ₂ C@CNT	0.25	This work
Ru@C ₂ N	1.9	Nat. Nanotechnol. 12, 441-446 (2017)
Ru@MWCNT	2.4	Nat. Commun. 11, 1278 (2020)
Pt NPs	2.82	Angew. Chem. Int. Ed. 58, 5432-5437 (2019)
PtNi-O/C	2.18	J. Am. Chem. Soc. 140, 9046-9050 (2018)
Pt (pc)	0.5	ECS Transactions. 50, 2163 (2013)
R-MoS ₂ @NF	2.27	Adv. Mater. 30, 1707105 (2018)
Mo ₂ C	0.12	Nat. Commun. 10, 1217 (2019)
C-MoS ₂	1.28	Nat. Commun. 10, 1217 (2019)
Pt/MMC	3.2	Adv. Funct. Mater. 29, 1901217 (2019)

Manuscript Revision: The corresponding sentence “At the same time, the exchange current densities of Ru-Mo₂C@CNT, Pt/C, Ru@CNT and Mo₂C@CNT are extrapolated by Tafel to be 4.4, 2.2, 2.0 and 0.25

mA cm⁻², respectively. Obviously, the intrinsic catalytic activity of Ru-Mo₂C@CNT with a small-sized Ru-Mo₂C heterostructure is the best, and it exhibits the best HER performance in alkaline media.” has been added in the revised manuscript (please see **14-17 lines** (the red-label part) of **Page 9** in the revised manuscript). And the corresponding sentence “The exchange current densities of Ru-Co₃C@CNT and Ru-Cr₂₃C₆@CNT are also extrapolated from Tafel, which are 2.9 and 3.8 mA cm⁻², respectively.” has also been added in the revised manuscript (please see **12-13 lines** (the red-label part) of **Page 11** in the revised manuscript).

The **Figure R3** has been corrected in the Manuscript (please see **Supplementary Figure 28c**). The **Table R1** has been given in the Supporting Information (please see **Supplementary Table 4**). The references (*ACS Catal.* 2016, 6, 1929-1941; *ChemSusChem*, 2018, 11, 2388-2401) have been added in the revised manuscript as **Reference 56** and **57** on **page 23**.

Question 4: Furthermore, the determination of the ECSA via underpotential deposition of Cu is completely wrong. In Figure S16 and S28, the anodic peaks of a Cu monolayer appear at the same potentials like the redox potential of Cu/Cu²⁺. The authors mainly observed a bulk dissolution of metallic Cu and this fact explains the high ECSA values.

Author Reply 4: Thanks for your valuable suggestion. We retested the Cu UPD (**Figures R4** and **R6**) of the catalysts and corrected the ECSA. The ECSA values are calculated by formulas $ECSA (cm^2_{metal}/g_{metal}) = \frac{Q_{Cu}}{M_{metal} \times 420 \mu C cm^{-2}}$. As shown in **Figures R5** and **R7**, after recalculation, the ECSA of Ru-Mo₂C@CNT is 97.6 m²g⁻¹_{Ru}, this result indicates that the electrochemically active area of the catalyst is larger than some other materials reported in the literature.

Figure R4. (a) Copper UPD in 0.5 M H_2SO_4 in the (I) absence and (II- X) presence of 5 mM CuSO_4 on Pt/C. For II-X, the electrode was polarized at 0.3V, 0.29V, 0.28V, 0.27V, 0.26V, 0.25V, 0.24V, 0.23V and 0.22V for 100 s to form the UPD layers, respectively. (b) Copper UPD in 0.5 M H_2SO_4 in the (I, II) absence and (III) presence of 5 mM CuSO_4 on Pt/C. For II and III, the electrode was polarized at 0.26 V for 100 s to form the UPD layer. (c) Copper UPD in 0.5 M H_2SO_4 in the (I) absence and (II-VII) presence of 5 mM CuSO_4 on Ru-Mo₂C@CNT. For II-VII, the electrode was polarized at 0.26V, 0.25V, 0.24V, 0.235V, 0.23V and 0.225V for 100 s to form the UPD layers, respectively. Copper UPD in 0.5 M H_2SO_4 in the absence and presence of 5 mM CuSO_4 on Ru-Mo₂C@CNT. The electrode was polarized at 0.23 V for 100 s to form the UPD layer. (d) Copper UPD in 0.5 M H_2SO_4 in the (I, II) absence and (III) presence of 5 mM CuSO_4 on Ru-Mo₂C@CNT. For II and III, the electrode was polarized at 0.23 V for 100 s to form the UPD layer.

Figure R5. (a) Estimation of the ECSA of Ru-Mo₂C@CNT, Pt/C, Ru@CNT and Mo₂C@CNT.

Figure R6. Copper UPD in 0.5 M H₂SO₄ in the (I, II) absence and (III) presence of 5 mM CuSO₄ on (a) Ru-Cr₂₃C₆@CNT and (b) Ru-Co₃C@CNT. For II and III, the electrode was polarized at 0.23 V for 100 s to form the UPD layer.

Figure R7. (a) Estimation of the ECSA of Ru-Mo₂C@CNT, Ru-Cr₂₃C₆@CNT and Ru-Co₃C@CNT.

Manuscript Revision: The corresponding sentence “The ECSA of Ru-Mo₂C@CNT is 97.6 m²g⁻¹_{Ru}, which is larger than Pt/C (73.8 m²g⁻¹), Ru@CNT (69.1 m²g⁻¹) and Mo₂C@CNT (42.8 m²g⁻¹) (Supplementary Figure 18).” has been corrected in the revised manuscript (please see 21-23 lines (the red-label part) of Page 9 in the revised manuscript). The corresponding sentence “When evaluating HER electrocatalysts, the turnover frequency (TOF) and the overpotential at 10 mA cm⁻² respectively reveal the intrinsic activity of the catalyst and the potential for practical applications. According to the estimated number of active sites, the TOF value of each active site of Ru-Mo₂C@CNT, Pt/C, Ru@CNT and Mo₂C@CNT in alkaline electrolyte was calculated (Supplementary Figures 18).” has been corrected in the revised manuscript (please see 2-5 lines (the red-label part) of Page 10 in the revised manuscript). And the corresponding sentence “By calculation, the ECSA of Ru-Co₃C@CNT and Ru-Cr₂₃C₆@CNT catalysts were 76.2 and 88.1 m²g⁻¹_{Ru}, respectively (Supplementary Figure 30c). The TOF was obtained by the same method, as shown in Supplementary Figure 28d, Ru-Co₃C@CNT, Ru-Cr₂₃C₆@CNT and Ru-Mo₂C@CNT all showed a large TOF value, which was 10.3, 9.2 and 21.9 s⁻¹ under the overpotential of 100 mV, respectively.” have been corrected in the revised manuscript (please see 15-19 lines (the red-label part) of Page 11 in the revised manuscript).

The Figure R4, Figure R5, Figure R6 and Figure R7 have been given in the Supporting Information, respectively. (please see Supplementary Figure 17, Supplementary Figure 18 and Supplementary Figure 30.)

Question 5: In Figure S32, the Nyquist plot starts for all materials nearly at $Z'=0$ Ohm. This is an observation which can not be explained by the presented data.

Author Reply 5: Thanks for your kind suggestion. After retesting, the Nyquist diagrams of all materials almost start around $Z'=2.1$ ohms, which indicates that the solution resistance (or contact resistance) R_s is small.

Figure R7. EIS Nyquist plots of Mo₂C@CNT, Ru@CNT and Ru-Mo₂C@CNT.

Figure R8. EIS Nyquist plots of Ru-Cr₂₃C₆@CNT and Ru-Co₃C@CNT and Ru-Mo₂C@CNT catalysts.

Manuscript Revision: We changed the original sentences to “The electrochemical impedance spectroscopy (EIS) fitting results showed that the charge transfer resistance of Ru-Mo₂C@CNT (21.3 Ω) is smaller than that of Ru@CNT (26.0 Ω) and Mo₂C@CNT (30.6 Ω) (Supplementary Figure 21).” and

“Besides, the electrochemical impedance spectroscopy (EIS) (**Supplementary Figure 32**) showed that both Ru-CO₃C@CNT (24.1 Ω) and Ru-Cr₂₃C₆@CNT (22.4 Ω) also had smaller impedance values, indicating that such materials had higher charge transfer rates and easier HER reaction kinetics.” in the revised manuscript (please see **13-15 lines** (the red-label part) of **Page 10** and **1-3 lines** (the red-label part) of **Page 12** in the revised manuscript).

The **Figure R7** and **Figure R8** have been corrected in the Supporting Information, respectively (please see **Supplementary Figure 21** and **Supplementary Figure 32**).

Question 6: It is also frustrated to see that the authors did not make enough effort to measure their materials in a proper manner. The Raman, XRD, EDX and other data are presented in a poor quality.

Author Reply 6: Thanks for reviewer’s valuable comment and suggestion. As shown in the figures below, we re-provide a clearer Raman graph (**Figures R9**), XRD images of MWCNT (**Figures R10**), Ru@CNT (**Figures R11**), Mo₂C@CNT (**Figures R12**), EDX mapping (**Figures R13, R14**) and the TEM images of Ru-Mo₂C@CNT (**Figures R15**) and Mo₂C@CNT (**Figures R16**).

Figure R9. Raman spectra of Ru@CNT, Mo₂C@CNT and Ru-Mo₂C@CNT catalysts.

Figure R10. XRD image of MWCNT.

Figure R11. XRD image of Ru@CNT.

Figure R12. XRD image of Mo₂C@CNT.

Figure R13. HAADF-STEM image and corresponding EDX maps of Ru-Mo₂C@CNT for Ru, Mo and Ru+Mo, respectively.

Figure R14. EDX spectrum of Ru-Mo₂C@CNT.

Figure R15. (a, b) TEM images of Ru-Mo₂C@CNT.

Figure R16. TEM image of Mo₂C@CNT.

Manuscript Revision: We changed the original sentence to “It can be seen from **Supplementary Figure 14** that the intensity ratio of the D band and the G band (I_D/I_G) of the catalyst Ru-Mo₂C@CNT is 1.45, which was significantly higher than the intensity ratio of the control samples Ru@CNT (0.91) and Mo₂C@CNT (1.23).” in the revised manuscript (please see **10-12 lines** (the red-label part) of **Page 8** in the revised manuscript).

The **Figure R9** has been given in the Manuscript and Supporting Information (please see **Supplementary Figure 15**). The **Figure R10**, **Figure R11** and **Figure R12** have been given in the Manuscript and Supporting Information, respectively (please see **Supplementary Figure 2**, **Supplementary Figure 9** and **Supplementary Figure 13**). The **Figure R13** and **Figure R14** have been given in the Manuscript and Supporting Information, respectively (please see **Figure 1e** and **Supplementary Figure 7**). The **Figure R15** and **Figure R16** have been given in the Manuscript and Supporting Information, respectively (please see **Supplementary Figure 6(c, d)** and **Supplementary Figure 11b**).

To Reviewer 3

General Comment: Electrocatalytic hydrogen evolution reaction (HER) by splitting water has become an effective method for the sustainable production of H₂. It is highly desirable and imperative to develop new HER electrocatalysts with low-cost and high-performance. Herein, the authors reported a simple, fast and solvent-free microwave pyrolysis method for the synthesis of ultra-small (3.5 nm) Ru-MxC@CNT (M=Mo, Co, Cr) catalyst with heterogeneous structure and strong metal-support interaction in one step. The fabricated Ru-Mo₂C@CNT catalyst exhibits a low overpotential of 15 mV at a current density of 10

mA cm^{-2} , and exhibits a large TOF value up to 57.8 s^{-1} under an overpotential of 100 mV. This paper is interesting, and I recommend this paper can be accepted after following revisions.

Author Reply: Thank you for the valuable confirmation of our work. We highly appreciate your efforts in reviewing our work and giving valuable comments. Based on your valuable comments, we have supplied the detailed.

Question 1: The author claims that the synthesis of Ru-Mo₂C@CNT, however, in Figure 1, only one Ru-Mo₂C nanoparticle is seen in Figure 1c, indicating that the Ru-Mo₂C nanoparticles are not ubiquitous on CNT. So the TEM image with more Ru-Mo₂C nanoparticles should be provided. This is crucial for this paper.

Author Reply 1: Thanks for your helpful comment and suggestion. To show that the presence of Ru-Mo₂C nanoparticles on CNTs is universal, TEM images with more Ru-Mo₂C nanoparticles are added and shown in **Figure R1d**.

Figure R1. TEM image of Ru-Mo₂C@CNT.

Manuscript Revision: The **Figure R1** has been given in the Supporting Information (please see **Supplementary Figure 6d**.)

Question 2: For Ru-Mo₂C@CNT, how about the content of Ru or Mo₂C on the catalytic activity and stability for HER? The authors should provide more information.

Author Reply 2: Thanks for reviewer's valuable suggestion. After experimental testing, we found that when the content ratio of Ru and Mo₂C is 2:1, Ru-Mo₂C@CNT has the best HER catalytic activity (**Figure R2 a-b**). Although the content ratio of Ru and Mo₂C is different, these catalysts all have excellent electrochemical stability (**Figure R 2c**).

Figure R2. (a) Polarization curves of the Ru-Mo₂C@MWCNT catalysts prepared under different Ru: Mo₂C ratios of 1:1, 2:1 and 4:1 in 1.0 M KOH solution. (b) Corresponding Tafel slopes in 1.0 M KOH solution. (c) Current-time (i-t) stability curves up to 12 h duration of different Ru: Mo₂C ratios of 1:1, 2:1 and 4:1 were recorded in 1.0 M KOH solutions.

Manuscript Revision: We changed the original sentence to “Based on the 1:1 ratio of metal and support, the best content ratio of Ru: Mo₂C was explored. Among them, the initial reactants Ru₃(CO)₁₂ and Mo(CO)₆ with mass ratios of 1:2, 1:1, and 2:1 were placed in a microwave oven for reaction.” (please see **1-2 lines** (the red-label part) of **Page 6**) and “Electrochemical hydrogen evolution tests were performed on three samples of different proportions in 1.0 M KOH solution, and we found that the electrochemical performance was the best when the Ru:Mo₂C element content ratio was 2:1 (**Supplementary Figure 5**).” (please see **4-7 lines** (the red-label part) of **Page 6**) in the revised manuscript.

The **Figure R2** has been given in the Supporting Information (please see **Supplementary Figure 5**.)

Question 3: How about the long-term durability of Ru-Mo₂C@CNT for HER in alkaline media? As shown in Figure 4f, the test time of durability is too short, and the authors should measure the durability of HER for at least 50 hours.

Author Reply 3: Thanks for reviewer’s valuable comment and suggestion. We have retested the electrochemical stability of the catalyst in 1.0 M KOH electrolyte for 100 hours, and the results also prove that Ru-Mo₂C@CNT has good long-term durability.

Figure R3. Current-time (i-t) stability curves up to 100 h duration of Ru-Co₃C@CNT, Ru-Cr₂₃C₆@CNT and Ru-Mo₂C@CNT were recorded in 1.0 M KOH solutions.

Manuscript Revision: We changed the original sentence to “The current-time (i-t) test (Figure 4f) showed that the current density remained almost constant for 100 hours.” in the revised manuscript (please see 6-7 lines of Page 12 in the revised manuscript).

The **Figure R3** has been given in the Manuscript (please see **Supplementary Figure 28f.**)

Question 4: How do you prove the structure stability of Ru-Mo₂C@CNT for long-term durability of HER ? Please provide some evidences for structure stability of Ru-Mo₂C@CNT after durability test.

Author Reply 4: Thanks for your helpful suggestion. In the supporting information of the article, we provide the SEM, TEM and XRD data of the Ru-Mo₂C@CNT catalyst after the HER long-term stability test to prove that the catalyst has good structural stability. In order to further prove the excellent stability of this catalyst, we have added its XPS test. As shown in **Figure R4**, after the HER reaction, the peaks of each element basically did not change.

Figure R4. XPS pattern of Ru-Mo₂C@CNT after stability test.

Manuscript Revision: We changed the original sentence to “In addition, the SEM and TEM images of Ru-Mo₂C@CNT after the long-term stability test showed no change in the morphology of the material, and the XRD and XPS images further reflected that the structure of this material did not change (Supplementary Figures 35-36).” in the revised manuscript (please see **10-12 lines of Page 13** in the revised manuscript).

The **Figure R4** has been given in the Supporting Information (please see **Supplementary Figure 36**.)

Question 5: The loading of Ru-Mo₂C on CNTs should be provided for Figure 4, and the effect of the loading of Ru-Mo₂C on CNTs on the catalytic activity and stability should be studied.

Author Reply 5: Thanks for your valuable suggestions. For the catalyst in **Figure R5**, the ratio of Ru-Mo₂C to CNT is 1:1, that is, 5 mg of metal elements are loaded on 5 mg of CNT. In addition, we studied the effects of different Ru-Mo₂C loads on the catalytic activity and stability of CNTs, as shown in Figure R5. The results showed that the catalyst had the best catalytic activity when the contents of Ru-Mo₂C and CNTs were the same. When the content of Ru-Mo₂C was lower than that of CNTs, the

number of active sites decreased and the catalytic activity decreased. When the content of Ru-Mo₂C was higher than that of CNTs, too much load blocked part of the active sites, and the catalytic activity decreased slightly. Therefore, only when the content of Ru-Mo₂C was the same as that of CNTs, the active sites were most fully exposed and the catalytic activity was the best. In addition, the catalysts with different Ru-Mo₂C loading loads have good electrochemical stability.

Figure R5. (a) Polarization curves of the Ru-Mo₂C@MWCNT catalysts prepared under different Ru-Mo₂C@CNT ratios of 1:2, 1:1 and 3:2 in 1.0 M KOH solution. (b) Corresponding tafel slopes in 1.0 M KOH solution. (c) Current-time (i-t) stability curves up to 12 h duration of different Ru-Mo₂C@CNT ratios of 1:2, 1:1 and 3:2 were recorded in 1.0 M KOH solutions.

Manuscript Revision: The corresponding sentence “In order to confirm the optimal metal to support ratio, we first explored the HER catalytic activity of catalysts with different Ru-Mo₂C:CNT ratios. As shown in **Supplementary Figure 4**, when the ratio of metal to support is 1:1, the catalytic performance is the best.” has been added in the revised manuscript (please see **20-22 lines** (the red-label part) of **Page 5** in the revised manuscript).

The **Figure R5** has been given in the Supporting Information (please see **Supplementary Figure 5**.)

Question 6: For the data shown in Figure 4g, to maintain the current density at 500 mA cm⁻², the overpotential should be provided.

Author Reply 6: Thanks for your helpful suggestion. As shown in **Figure R6**, we have supplemented the chronopotentiometric test of the Ru-Mo₂C@CNT catalyst supported on Ni foam at 500 mA cm⁻². The results also prove that the Ru-Mo₂C@CNT material has excellent electrochemical stability required by the industry.

Figure R6. The chronopotentiometric curve of the Ru-Mo₂C@CNT electrode tested at a constant current density of 500 mA cm⁻² for 500 h.

Manuscript Revision: The corresponding sentence “In addition, as shown in **Supplementary Figure 33**, the chronopotentiometric curve of the Ru-Mo₂C@CNT electrode was tested at a constant current density of 500 mA cm⁻² for 500 h. And this result further prov that the Ru-Mo₂C@CNT material has excellent stability.” has been added in the revised manuscript (please see **6-8 lines** (the red-label part) of **Page 13** in the revised manuscript).

The **Figure R6** has been given in the Supporting Information (please see **Supplementary Figure 33**.)

Question 7: Why does Ru-Mo₂C have higher catalytic performance than the other Ru-Co₃C@CNT, Ru-Cr₂₃C₆@CNT? Please provide some explains in the paper.

Author Reply 7: Thanks for your kind comments. According to the analysis of the peaks in the 3p orbital of Ru in the XPS data of Ru-Mo₂C@CNT, Ru-Co₃C@CNT and Ru-Cr₂₃C₆@CNT, it can be obtained by adding the Mo element, the binding energy of Ru element moved to a higher position by nearly 0.31 eV, and after adding Co and Cr elements, the binding energy moved 0.46 eV and 0.19 eV, respectively. After a lot of investigation literature, we found that proper electron transfer on the catalyst surface can promote the electrocatalytic performance of the material (Angew. Chem. Int. Ed. 2021, 60, 4110-4116; Angew. Chem. Int. Ed. 2014, 53, 122-126.). For example, the literature [Adv. Mater. 2020, 32, 2005433] shows that the Ru 3p binding energy of RuMo nanoalloy-embedded 2D porous carbon (2DPC-RuMo) nanosheets slightly shifts to higher binding energy ~0.3 eV, the electronic structure is adjusted to the most suitable position, which further indicates that proper electron transfer is beneficial to promote alkaline

HER Catalytic activity. In addition, the surface of the 2DPC-RuMo catalyst has the best Gibbs free energy of intermediates corresponding to the chemically adsorbed H*. Moreover, after a lot of research literature, we found that both the doping of Mo atoms [Adv. Mater. 2020, 32, 2005433] and the formation of Mo₂C [Adv. Funct. Mater. 2019, 29, 1901217] can provide an easier water dissociation process. Therefore, compared with Ru-Co₃C@CNT and Ru-Cr₂₃C₆@CNT, Ru-Mo₂C@CNT has greater advantages in regulating electronic structure and promoting water dissociation, and therefore its electrocatalytic HER activity is higher.

Question 8: Some relevant references about hydrogen evolution electrocatalysis may be considered to be cited, such as Angew. Chem. Int. Ed. 2017, 56, 2960; Angew. Chem. Int. Ed. 2017, 56, 8120; J. Am. Chem. Soc. 2018, 140, 5118.

Author Reply 8: Thanks for your comments. The references (*Angew. Chem. Int. Ed.* 2017, 56, 2960; *Angew. Chem. Int. Ed.* 2017, 56, 8120; *J. Am. Chem. Soc.* 2018, 140, 5118.) have been added in the revised manuscript as **Reference 9, 15 and 6 on Page 18 and page 19**.

Question 9: For supplementary Figure 21, the TEM image of Ru-Co₃C@CNT does not show the existence of heterojunction, please provide another TEM.

Author Reply 9: Thanks for your comments. In order to prove the existence of heterojunction in Ru-Co₃C@CNT catalyst, we added HRTEM images with heterojunction, and further proved the successful preparation of Ru-Co₃C@CNT catalyst (**Figure R7**). At the same time, we also added HRTEM images of Ru-Cr₂₃C₆@CNT with heterojunction (**Figure R8**).

Figure R7. HRTEM image of Ru-Co₃C@CNT.

Figure R8. HRTEM image of Ru-Cr₂₃C₆@CNT.

Manuscript Revision: The **Figure R7** and **Figure R8** have been given in the Supporting Information, respectively (please see **Supplementary Figure 22d** and **Supplementary Figure 25d**).

In the end, we would like to express our thanks to the precious time of the reviewers and the editor. We sincerely wish that our point-to-point response and the revised manuscript can address your concerns and satisfy your requirements for publications. We would be grateful if we have the chance to share our work with readers of *Nature Communications*.

Sincerely yours,

Lei Wang

REVIEWER COMMENTS

Reviewer #1 (Remarks to the Author):

Thank you for considering the comments, I also thank reviewers 2 and 3 for bringing up very important observations about this paper. This paper has brought insight into a novel material for HER, and showed how it fared against similar materials.

With the corrections applied, I believe this paper is suitable for publishing.

Reviewer #2 (Remarks to the Author):

The revised manuscript reported by Wu et al. addressed most of the comments by the reviewer. The presented reference material (Pt/C) is now in the well-known range.

Page 9, line 12/13: Some statements are not correct in view of the referee. The referee cannot agree with the proposed rate determining step. The following statement needs revision or more explanation about the Volmer-Tafel versus Volmer-Heyrovsky. Why is the H₂ desorption an electrochemical step?

The preparation of MWCNT in the Methods Section is removed in the present version. Please explain. In the view of catalytic improvement by support / metal interaction this information should not be missing.

Furthermore, it's hard for the readership to figure out which the real catalyst loading (e.g. wt.% metal) was applied. Please calculate the values and provide the information within the manuscript.

The referee is wondering about the minimum effort to analyze the data by using Butler-Volmer equation in a careful way. Numerous calculations are not clear or based on highly distributed values. Please show the Butler-Volmer plots, discuss the transfer coefficient for HER/HOR and explain the calculated values.

Reviewer #3 (Remarks to the Author):

This paper is well revised, and now it can be accepted as it is.

Lei Wang

College of Chemistry and Molecular Engineering

Qingdao University of Science and Technology, Qingdao 266042, P. R. China.

E-mail: inorchemwl@126.com

May. 24, 2021

Dear Editor,

We highly appreciate your kind consideration and review on our paper entitled “*Solvent-free microwave synthesis of ultra-small Ru-Mo₂C@CNT with strong metal-support interaction for industrial alkaline hydrogen evolution reaction*” (NCOMMS-20-48819). We have carefully revised the manuscript and responded all the raised valuable comments by three reviewers. We have also highlighted the changes in manuscript and supporting information with red color. Thanks to the valuable suggestions, our manuscript could be significantly improved. We response to the reviewers’ comments point by point and highlight the changes in the revised manuscript. The list of changes and our responses to three reviewers’ comments are provided as follows.

Reply to Reviewers' Comments

Dear Reviewers,

Thank you for your precious time to constructive comments on our manuscript titled “**Solvent-free microwave synthesis of ultra-small Ru-Mo₂C@CNT with strong metal-support interaction for industrial alkaline hydrogen evolution reaction**” (Manuscript ID: NCOMMS-20-48819) for *Nature Communications*. We sincerely appreciate your opinions and confirmation of our work. Accordingly, we have supplied the corresponding response and revision based on the comments. We sincerely hope that our responses will fully address your concerns about our work.

To Reviewer 1:

General Comment: Thank you for considering the comments, I also thank reviewers 2 and 3 for bringing up very important observations about this paper. This paper has brought insight into a novel material for HER, and showed how it fared against similar materials.

With the corrections applied, I believe this paper is suitable for publishing.

Author Reply: Thank you for your precious support and appreciation of our work. Your kind comments are very supportive to our present work and our future works in this field. We sincerely hope that our work will deliver a novel research to the electrocatalyst design in the future.

To Reviewer 2:

General Comment: The revised manuscript reported by Wu et al. addressed most of the comments by the reviewer. The presented reference material (Pt/C) is now in the well-known range.

Author Reply: Thank you for the valuable confirmation of our work. We highly appreciate your efforts in reviewing our work and giving valuable comments. Based on your valuable comments, we have supplied the detailed.

Question 1: Page 9, line 12/13: Some statements are not correct in view of the referee. The referee cannot agree with the proposed rate determining step. The following statement needs revision or more explanation about the Volmer-Tafel versus Volmer-Heyrovsky. Why is the H₂ desorption an electrochemical step?

Author Reply 1: Thank you for your helpful comment. According to (Angew. Chem. Int. Ed. 2021, 60, 4110-4116), (Adv. Energy Mater. 2019, 9, 1900931) and (Nat. Nanotechnol. 2017, 12, 441-446), it can be

obtained that when the Tafel slope is 26 mV dec^{-1} , following the Volmer-Tafel step. Similarly, the literature (Nat. Nanotechnol. 2017, 12, 441-446 and Nat. Commun. 2017, 8, 14969) suggested that when HER follows the Volmer-Tafel mechanism, the electrochemical desorption of H_2 is the rate determining step of the HER process.

Question 2: The preparation of MWCNT in the Methods Section is removed in the present version. Please explain. In the view of catalytic improvement by support / metal interaction this information should not be missing.

Author Reply 2: Thanks for your valuable comment. I have added its specific experimental steps to the manuscript.

Manuscript Revision: The corresponding sentence “**Preparation of MWCNT.** Disperse 50 mg MWCNT powder in a mixed solution with a concentration of H_2SO_4 : HNO_3 =3:1 for ultrasonic treatment for 1-2 h, and then expose it to 1 M HCl for ultrasonic treatment for 30 minutes. Finally, filter the acidified CNT, wash it with ionized water until $\text{pH} = 7$, and dry it at 60°C for 12 h.” has been added in the revised manuscript (please see **1-5 lines** (the red-label part) of **Page 15** in the revised manuscript).

Question 3: Furthermore, it’s hard for the readership to figure out which the real catalyst loading (e.g. wt.% metal) was applied. Please calculate the values and provide the information within the manuscript.

Author Reply 3: Thanks for reviewer’s valuable comment and suggestion. In order to enable readers to get a clearer view of the metal content in the catalyst, we have carried out a detailed description of the catalyst content in the manuscript.

Manuscript Revision: The corresponding sentence “And it was estimated by ICP-AES results that the loading of Ru-Mo₂C in the catalyst Ru-Mo₂C@CNT is about 19 wt%.” has been added in the revised manuscript (please see **7-8 lines** (the red-label part) of **Page 6** in the revised manuscript). The corresponding sentence “(catalyst loading is 0.14 mg cm^{-2} , equals to a Ru-Mo₂C loading of ca. 0.03 mg cm^{-2})” has also been added in the revised manuscript (please see **20 line** (the red-label part) of **Page 8** in the revised manuscript). And the corresponding sentence “(When the Ru-Mo₂C loading amount in the catalyst supported on NF is 0.95 mg cm^{-2} , the catalyst Ru-Mo₂C@CNT achieved the industrial current densities of 500 mA cm^{-2} and 1000 mA cm^{-2} at low overpotentials of 56 mV and 78 mV (Figure 4d), respectively.)” has also been added in the revised manuscript (please see **3-4 lines** (the red-label part) of **Page 13** in the revised manuscript).

Question 4: The referee is wondering about the minimum effort to analyze the data by using Butler-Volmer equation in a careful way. Numerous calculations are not clear or based on highly distributed values. Please show the Butler-Volmer plots, discuss the transfer coefficient for HER/HOR and explain the calculated values.

Author Reply 4: Thanks for your valuable suggestion. We supplemented the Butler-Volmer plots of HER/HOR, and obtained the symmetry factor and exchange current density of Ru-Mo₂C@CNT and commercial Pt/C according to the Koutecky-Levich equation and Butler-Volmer equation. As shown in R1 and R2, the transfer coefficients of Ru-Mo₂C@CNT and commercial Pt/C are 0.45 (**Figure R1a**) and 0.47 (**Figure R2a**), respectively. All curves can be installed in the range of 0.4-0.6, indicating that the branches of HOR and HER have good symmetry. The j_0 values obtained from linear fitting of micropolarization regions (**Figure R1b and R2b**) are consistent with the values of j_0 obtained from Butler-Volmer fitting. And, it is almost consistent with the results obtained by tafel slope extrapolation before.

Figure R1. (a) HOR/HER Tafel plots of the kinetic current density on Ru-Mo₂C@CNT in H₂-saturated 1.0 M KOH. The solid line indicate the Butler-Volmer fitting. (b) Micropolarization regions (-10 mV to 10 mV) of Ru-Mo₂C@CNT. In the Micropolarization region, Butler-Volmer equation can be simplified to $j_0 = \frac{j RT}{\eta F}$, where j is the measured current density, η is the overpotential, R is the universal gas constant, T is the temperature, and F is Faraday's constant. Therefore, the exchange current density (j_0) can be obtained from the slope of the linear fitting of j - η curve in micropolarization regions.

Figure R2. (a) HOR/HER Tafel plots of the kinetic current density on Pt/C in H₂-saturated 1.0 M KOH. The solid line indicate the Butler-Volmer fitting. (b) Micropolarization regions (-10 mV to 10 mV) of Pt/C.

Manuscript Revision: The corresponding sentence “Exchange current density (J_0) is another important parameter reflecting translation kinetics, which can provide the internal electron transfer rate between the electrode and the catalyst surface. The exchange current density (J_0) of the catalyst was extracted from the linear fitting of the micro-polarization region (-10 to 10 mV). The exchange current density of Ru-Mo₂C@CNT was 4.3 mA cm⁻² (**Figure 3c and Supplementary Figure 17b**), which was better than commercial Pt/C (2.3 mA cm⁻²) (**Figure 3c and Supplementary Figure 18b**). The exchange current densities of Ru@CNT and Mo₂C@CNT were 2.0 and 0.28 mA cm⁻², respectively (**Figure 3c**), which were slightly lower than the J_0 of the Pt/C catalyst. These values are in good agreement with the fitting results of the Butler-Volmer equation in the Tafel region (**Supplementary Figure 17a and Supplementary Figure 18a**).” has been corrected in the revised manuscript (please see **16-23 lines** (the red-label part) of **Page 9** in the revised manuscript). The corresponding sentence “The exchange current densities of Ru-Co₃C@CNT and Ru-Cr₂₃C₆@CNT are also studied from linear fitting of micropolarization regions, which are 2.7 and 3.6 mA cm⁻², respectively (**Supplementary Figure 30c**).” has been corrected in the revised manuscript (please see **21-23 lines** (the red-label part) of **Page 11** in the

revised manuscript). And the corresponding HER/HOR experiments and calculation methods are added to the “Methods” section of the manuscript (please see **7-20 lines** (the red-label part) of **Page 17** and **1-3 lines** (the red-label part) of **Page 18** in the revised manuscript). The specific content is as follows:

Electrochemical hydrogen production/oxidation (HER/HOR) reaction test At this time, a disk electrode (RDE area: 0.196 cm²) was used as the working electrode. In 1.0 M KOH electrolyte saturated with N₂, the CV scan was performed at a scan rate of 100 mV/s from 0.05 V to 1.10 V vs. RHE until it stabilized. The HER/HOR test was performed by linear sweep voltammetry (LSV) in a 1.0 M KOH solution saturated with H₂ using a sweep rate of 10 mV/s, a rotation speed of 1,600 rpm, and all data were iR corrected. The exchange current density (J_0) and the symmetry factor (α) are obtained by fitting kinetic current density (j_k) at small current density region into the Butler-Volmer equation as follows:

$$j_k = j_0 \left(e^{\frac{\alpha F \eta}{RT}} - e^{-\frac{(\alpha-1)F\eta}{RT}} \right)$$

The current density (j) obtained from the working electrode is the sum of two currents: dynamic current density (j_k) and diffusion current density (j_d). The dynamic current density (j_k) is derived from the following Koutecky-Levich equation:

$$\frac{1}{j} = \frac{1}{j_k} + \frac{1}{j_d}$$

in which j_d obeys the Levich equation:

$$j_d = 0.62nFD^{2/3}\nu^{-1/6}C_0\omega^{1/2}$$

in which F is the Faraday constant (96 485 C mol⁻¹), n is the number of electrons involved in the oxidation reaction, C_0 is the H₂ concentration in solution, D is the diffusion coefficient of the reactant (cm²s⁻¹), ν is the viscosity of the electrolyte (cm²s⁻¹), and ω is the rotation speed (rpm).

The **Figure R1** and **Figure R2** have been given in the Supporting Information, respectively. (please see

Supplementary Figure 17 and Supplementary Figure 18.) The references (Adv. Funct. Mater. 2019, 29, 1901217; Nat. Commun., 2020, 11, 4789) have been added in the revised manuscript as **Reference 56** and **57** on **page 24**.

To Reviewer 3:

General Comment: This paper is well revised, and now it can be accepted as it is.

Author Reply: Thank you for your valuable comments on this article, and very grateful for your support for our work.

In the end, we would like to express our thanks to the precious time of the reviewers and the editor. We sincerely wish that our point-to-point response and the revised manuscript can address your concerns and satisfy your requirements for publications. We would be grateful if we have the chance to share our work with readers of *Nature Communications*.

Sincerely yours,

Lei Wang

REVIEWERS' COMMENTS

Reviewer #2 (Remarks to the Author):

The authors addressed the comments from the reviewer and improved their manuscript slightly.

In page 9, line 15/16. To make it clear. An electrochemical step involves an electron charge transfer, it can be also (de)coupled with a proton transfer.

The Tafel-Volmer mechanism evolves H₂ by combination of 2 H_{ad}, followed by a chemical desorption. Therefore, the desorption of molecular H₂ (forming by 2 H_{ad}) does not include an electron transfer.

The Tafel-Heyrovsky mechanism evolves H₂ by discharging of ion and atom (electrochemical desorption).

More basics can be read in the famous text book "Electrochemistry", Carl H. Hamann and Wolf Vielstich.

The authors have to rephrase their statements.

Reply to Reviewers' Comments

Dear Reviewer,

Thank you for your precious time to constructive comments on our manuscript titled “**Solvent-free microwave synthesis of ultra-small Ru-Mo₂C@CNT with strong metal-support interaction for industrial alkaline hydrogen evolution reaction**” (Manuscript ID: NCOMMS-20-48819) for *Nature Communications*. We sincerely appreciate your opinions and confirmation of our work.

To Reviewer 2:

General Comment: The authors addressed the comments from the reviewer and improved their manuscript slightly.

Author Reply: Thank you for the valuable confirmation of our work. We highly appreciate your efforts in reviewing our work and giving valuable comments. Based on your valuable comments, we have supplied the detailed.

Question 1: In page 9, line 15/16. To make it clear. An electrochemical step involves an electron charge transfer, it can be also (de)coupled with a proton transfer. The Tafel-Volmer mechanism evolves H₂ by combination of 2 H_{ad}, followed by a chemical desorption. Therefore, the desorption of molecular H₂ (forming by 2 H_{ad}) does not include an electron transfer. The Tafel-Heyrovsky mechanism evolves H₂ by discharging of ion and atom (electrochemical desorption). More basics can be read in the famous text book “Electrochemistry”, Carl H. Hamann and Wolf Vielstich. The authors have to rephrase their statements.

Author Reply 1: Thank you for your helpful comment. We have rewritten this statement into the manuscript.

Manuscript Revision: The corresponding sentence “In 1.0 M KOH solution, Ru-Mo₂C@CNT only showed a low Tafel slope of 26 mV dec⁻¹ (**Figure 3b**), which was much lower than Pt/C (44 mV dec⁻¹), Ru@CNT (76 mV dec⁻¹) and Mo₂C@CNT (197 mV dec⁻¹), which implies the Volmer-Tafel mechanism as the HER pathway, in which recombination of chemisorbed hydrogen atoms is the rate-determining step.” has been corrected in the revised manuscript (please see **8-9 lines** (the red-label part) of **Page 9** in the revised manuscript).

In the end, we would like to express our thanks to the precious time of the reviewer. We sincerely wish that our point-to-point response and the revised manuscript can address your concerns and satisfy your requirements for publications. We would be grateful if we have the chance to share our work with readers of *Nature Communications*.

Sincerely yours,

Lei Wang